The EMBO Journal (2013) 32, 1008–1022
www.embojournal.org

# PDK1 regulates VDJ recombination, cell-cycle exit and survival during B-cell development

Ram KC Venigalla[1,*], Victoria A McGuire[1,5], Rosemary Clarke[2,5], Janet C Patterson-Kane[3], Ayaz Najafov[1], Rachel Toth[1], Pierre C McCarthy[1], Frederick Simeons[4], Laste Stojanovski[4] and J Simon C Arthur[1,2,*]

[1]MRC Protein Phosphorylation Unit, Sir James Black Centre, College of Life Sciences, University of Dundee, Dundee, UK, [2]Division of Cell Signaling and Immunology, College of Life Sciences, University of Dundee, Dundee, UK, [3]Institute of Infection, Immunity and Inflammation, College of Medical, Veterinary and Life Sciences, University of Glasgow, Glasgow, UK and [4]Division of Biological Chemistry and Drug Discovery, College of Life Sciences, University of Dundee, Dundee, UK

**Phosphoinositide-dependent kinase-1 (PDK1) controls the activation of a subset of AGC kinases. Using a conditional knockout of PDK1 in haematopoietic cells, we demonstrate that PDK1 is essential for B cell development. B-cell progenitors lacking PDK1 arrested at the transition of pro-B to pre-B cells, due to a cell autonomous defect. Loss of PDK1 decreased the expression of the IgH chain in pro-B cells due to impaired recombination of the IgH distal variable segments, a process coordinated by the transcription factor Pax5. The expression of Pax5 in pre-B cells was decreased in PDK1 knockouts, which correlated with reduced expression of the Pax5 target genes *IRF4*, *IRF8* and *Aiolos*. As a result, Ccnd3 is upregulated in PDK1 knockout pre-B cells and they have an impaired ability to undergo cell-cycle arrest, a necessary event for Ig light chain rearrangement. Instead, these cells underwent apoptosis that correlated with diminished expression of the pro-survival gene Bcl2A1. Reintroduction of both Pax5 and Bcl2A1 together into PDK1 knockout pro-B cells restored their ability to differentiate *in vitro* into mature B cells.**

*The EMBO Journal* (2013) **32,** 1008–1022. doi:10.1038/emboj.2013.40; Published online 5 March 2013
*Subject Categories:* signal transduction; immunology
*Keywords*: B cell; Pax5; PDK1; preB; VDJ recombination

## Introduction

B cells play critical roles in the adaptive immune response and the formation of immunological memory. However if their development is not correctly regulated, B cells can give rise to serious pathologies. For instance, if self-reactive B cells are not deleted or repressed then they will lead to the development of autoimmunity, while mutations that activate B-cell proliferation can result in B-cell lymphomas or leukaemia (Nemazee, 2006; Herzog and Jumaa, 2012). It is therefore essential that B-cell development and proliferation are closely regulated and complex mechanisms have evolved to achieve this *in vivo*. How kinase signalling cascades contribute to this process is still not fully understood.

AGC kinases represent a family of closely related enzymes including PKC, Akt, PKA and RSK, and members of this family have been implicated in cellular proliferation, survival and differentiation. The activation of AGC kinases involves the phosphorylation of a specific residue in their activation (or 'T') loop as well as the phosphorylation of a second site in the C-terminus, referred to as the hydrophobic motif. Phosphoinositide-dependent kinase-1 (PDK1) is required for the phosphorylation of the activation loop of a subset of AGC kinases, including Akt, S6K, RSK and PKCs (Downward, 1998; Vanhaesebroeck and Alessi, 2000; Mora *et al*, 2004; Pearce *et al*, 2010). T loop phosphorylation is essential for the activation of Akt, S6K and RSK and consequently these kinases are inactive in cells lacking PDK1 (Williams *et al*, 2000). PDK1 therefore serves as a master regulator of a subgroup of AGC kinases. PDK1 is constitutively active in cells, and its ability to phosphorylate its substrates is regulated at the level of protein interaction. PDK1 regulates its targets by one of two mechanisms. For Akt, both Akt and PDK1 are recruited to the membrane by binding of their respective PH domains to PIP$_3$. This then permits PDK1 to phosphorylate the T loop (Thr308) of Akt, while the hydrophobic motif (Ser473) is phosphorylated by the mTORC2 complex (McManus *et al*, 2004; Pearce *et al*, 2010). Thus, the activation of Akt by PDK1 is dependent on PI-3 kinase activation, which catalyses the production of PIP$_3$ at the membrane. For other PDK1 substrates, the phosphorylation of the hydrophobic motif at the C-terminus of the AGC kinase domain is required to generate a docking site for PDK1, and significantly for these substrates PDK1 can act independently of PI-3 kinase signalling (Collins *et al*, 2003). Once docked, PDK1 then phosphorylates the T loop residue of the downstream kinase. The hydrophobic motif is phosphorylated by a distinct kinase, for instance mTORC1 for S6K and either autophosphorylation or MK2 for RSK (Zaru *et al*, 2007; Pearce *et al*, 2010).

Given the key role that PDK1 plays in signalling networks, it is an important kinase for several developmental processes. Knockout of PDK1 in mice results in embryonic lethality at E9.5 (Lawlor *et al*, 2002). Conditional knockout of PDK1 in muscle results in postnatal lethality, with the mice dying within 5–11 weeks of birth due to cardiac defects, while mice lacking PDK1 in the liver die within 4–16 weeks from liver failure (Mora *et al*, 2003, 2004). Conditional deletion in osteoclasts gives rise to skeletal defects that have a similarity to Rubinstein-Taybi syndrome (Shim *et al*, 2011).

*Corresponding authors. Present address. RKC Venigalla, Lymphocyte Signalling and Development, The Babraham Institute, Cambridge, CB22 3AQ, UK. Tel.: +44 1223 496467;
E-mail: ramkumar.venigalla@babraham.ac.uk or JSC Arthur, Division of Cell Signaling and Immunology, College of Life Sciences, University of Dundee, Dundee, UK. Tel.: +44 1382 384003; Fax: +44 1382 385783;
E-mail: j.s.c.arthur@dundee.ac.uk
[5]These authors contributed equally to this work.

The role of PDK1 in B-cell development however has not been directly addressed. Recent studies have started to dissect the roles of the PI-3 kinase signalling pathway in lymphocyte development (Okkenhaug and Vanhaesebroeck, 2003; Lazorchak *et al*, 2010; Ramadani *et al*, 2010; Baracho *et al*, 2011). For instance, mice with a conditional deletion of PI3Kα and an inactivating mutation in PI3Kδ in B cells exhibit a developmental block at the pro to pre-B stage (Ramadani *et al*, 2010). This could imply a role for PDK1 in B-cell development, however, PI-3 kinase signalling can also result in GEF and Tec kinase activation independently of PDK1 and Akt (Fruman, 2004; Wertheimer *et al*, 2012). Thus, it is also possible that the effects of PI-3 kinase inhibition in B-cell development occur via these Akt-independent pathways.

PDK1 has been shown to be important for the development of T cells (Hinton *et al*, 2004; Lee *et al*, 2005), loss of it results in T-cell developmental arrest at the DN3/4 stage in the thymus (Hinton *et al*, 2004). Deletion of PDK1 at the DN3 stage of development does not prevent recombination at the TCRβ locus, but does inhibit the progression of DN cells to the DP stage of development, resulting in very few T cells in the periphery (Hinton *et al*, 2004). T-cell development has also been shown to be sensitive to PDK1 levels; experiments using bone marrow from mice with a hypomorphic allele that results in ∼10% of the normal levels of PDK1 fail to competitively repopulate the T-cell pool in lethally irradiated mice. Interestingly in the same experiments, PDK1 had much less effect on B-cell repopulation (Hinton *et al*, 2004). In line with this, while knockout of Akt1 and 2 greatly perturbs T-cell development (Juntilla *et al*, 2007; Mao *et al*, 2007), it does not block B-cell development in the bone marrow (Calamito *et al*, 2009). The effect of Akt1/2 knockout in B cells was less severe that seen in the PI-3 kinase mutants (Ramadani *et al*, 2010). This difference may derive from several reasons, for example an Akt-independent function of PI-3 kinase in B-cell development or an upregulation of Akt3 activity in the Akt1/2 knockouts could explain these results. A further possibility is that the Akt knockout study used fetal liver cells to generate B cells, and it is possible that Akt may have less importance in embryonic rather than adult B cells. This suggests that PDK1 could be less critical for B-cell development; however, its role in this process has not been uncovered. To address this issue, we generated a conditional PDK1 knockout in the haematopoietic system using a Vav-Cre transgene. Interestingly, we observed that loss of PDK1 results in the arrest of B-cell development at the transition from pro-B to pre-B cells in the bone marrow. This was due to reduced Ig chain recombination that correlated with decreased expression of Pax5 and DNA ligase IV, proteins required for recombination events. In addition, PDK1 knockout pre-B cells underwent apoptosis due to a decrease in expression of the pro-survival gene Bcl2A1. The importance of Pax-5 and Bcl2A1 downstream of PDK1 was confirmed by the finding that reintroduction of both Pax5 and Bcl2A1 together into PDK1 knockout pre-B cells *in vitro* was required to restore their ability to differentiate into mature (IgD$^{+ve}$) B cells.

## Results

### Loss of PDK1 in haematopoietic cells blocks T and B cells but not myeloid cell development

To generate mice lacking PDK1 in haematopoietic cells, PDK1$^{fl/fl}$ mice were crossed to Vav-Cre transgenic mice, which express Cre early in haematopoietic development. Deletion of PDK1 was confirmed by qPCR of bone marrow, splenocytes and thymocytes. PDK1$^{fl/fl}$/Vav-Cre$^{+ve}$ were smaller than littermate controls (Supplementary Figure 1) and showed evidence of increased myeloid cell recruitment into the lung and liver (Supplementary Figure 2). In the lung, this was noted around and within arterial and venous walls, and there was significant associated arterial muscular hypertrophy. Despite the decreased body size, 6- to 24-week-old PDK1$^{fl/fl}$/Vav-Cre$^{+ve}$ mice had larger spleens relative to control genotypes (Figure 1A and B). However, while there was an increase in spleen size, following red blood cell lysis the splenocyte cell number was comparable between PDK1$^{fl/fl}$/Vav-Cre$^{+ve}$ knockout mice and control animals (Figure 1C). H&E staining revealed that the white pulp in PDK1$^{fl/fl}$/Vav-Cre$^{+ve}$ spleens was replaced by immature myeloid cells with increased numbers of granulocytes at various stages of maturity at the margins of this peri-arterial and peri-arteriolar tissue and throughout the red pulp. Increased numbers of siderophages were also noted. These observations indicated a defect in lymphocyte recruitment or development (Figure 1D). Consistent with the HE staining, FACS analysis of the splenocytes demonstrated that the PDK1-deficient spleens had an increased number of granulocytes and macrophages (Supplementary Figure 3). Normal numbers of conventional dendritic cells were found although the numbers of plasmacytoid dendritic cells was greatly reduced (Supplementary Figure 3). FACS analysis for TCRβ or B220-positive cells demonstrated that there were no clear mature B- or T-cell populations in the spleens of PDK1$^{fl/fl}$/Vav-Cre$^{+ve}$ mice (Figure 1F and E), in agreement with the absence of a defined white pulp (Figure 1D). This lack of T and B cells was not restricted to the spleen, as lymph nodes in the PDK1 knockout mice were small and contained no mature lymphocytes (Supplementary Figure 4). The lack of lymphocytes in the secondary immune organs could be explained by either a failure in development or migration. Analysis of the blood of PDK1$^{fl/fl}$/Vav-Cre$^{+ve}$ mice showed that there were no mature T or B cells present (Supplementary Figure 5), indicating that PDK1 was essential for either the development of T and B cells or their emigration from the lymphogenic organs. Deletion of PDK1 in the thymus at the DN3/4 stage of T-cell development has been shown to block T-cell development due to a decreased proliferation of DN4 cells and failure to upregulate CD4 and CD8 (Hinton *et al*, 2004). Deletion in the PDK1$^{fl/fl}$/VavCre$^{+ve}$ mice occurs in the bone marrow, earlier than the Lck-Cre used by Hinton *et al* (2004). Analysis of the thymi from PDK1$^{fl/fl}$/VavCre$^{+ve}$ mice demonstrated that there was an absence of CD4/CD8 DP cells and failure to upregulate the cell surface expression of TCRβ (Supplementary Figure 6). Development was arrested at the DN3 stage, however, expression of the intracellular TCRβ chain in DN3 cells was similar to that seen in wild-type cells (Supplementary Figure 6). Thus, PDK1 is essential for T-cell development, but not for recombination of the TCRβ locus. In T cells, PDK1 deletion has been correlated to decreased levels of the CD98 amino acid transporter and the transferrin receptor CD71, potentially resulting in metabolic stress as the DN4 cells proliferate (Kelly *et al*, 2007). In contrast, in B cells PDK1 knockout caused an increase in CD98 and CD71 levels in pro- and pre-B cells (Supplementary Figure 6), indicating that the roles of PDK1 may vary between T and B cells.

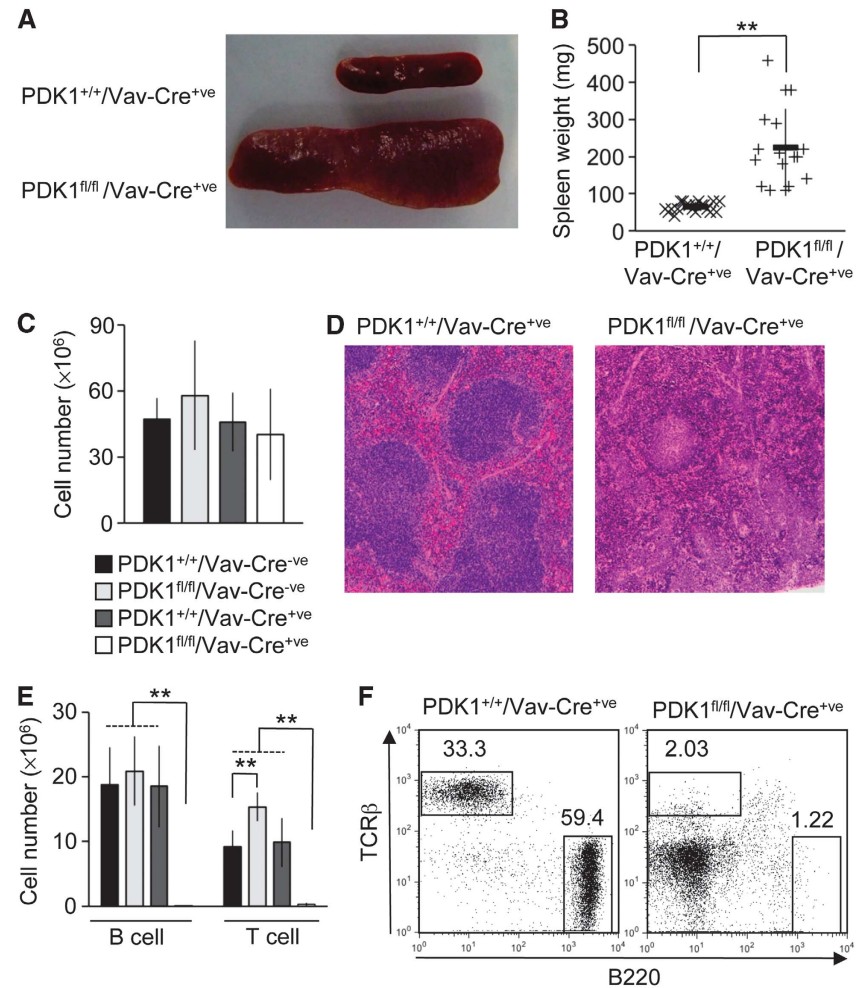

**Figure 1** PDK1 knockout in the haematopoietic system blocks the development of mature T and B cells. PDK1$^{fl/fl}$/Vav-Cre$^{+ve}$ mice were found to have an increased spleen size (**A**) and weight (**B**) relative to PDK1$^{+/+}$/Vav-Cre$^{+ve}$ control mice. Post lysis of red blood cells however spleen cell numbers were similar between the PDK1$^{fl/fl}$/Vav-Cre$^{+ve}$ mice ($n=15$) and PDK1$^{+/+}$/Vav-Cre$^{+ve}$ ($n=18$), PDK1$^{+/+}$/Vav-Cre$^{-ve}$ ($n=6$) and PDK1$^{fl/fl}$/Vav-Cre$^{-ve}$ ($n=6$) control genotypes (**C**). Error bars represent standard deviation. H&E staining of the spleens showed that PDK1 knockout resulted in a disruption of the white pulp (**D**). Analysis of T- and B-cell populations in the spleen by FACS (**E**, **F**) demonstrated that PDK1$^{fl/fl}$/Vav-Cre$^{+ve}$ mice ($n=7$) had negligible numbers of mature T and B cells relative to PDK1$^{+/+}$/Vav-Cre$^{+ve}$ ($n=6$), PDK1$^{+/+}$/Vav-Cre$^{-ve}$ ($n=6$) and PDK1$^{fl/fl}$/Vav-Cre$^{-ve}$ ($n=4$) control genotypes. Error bars represent the standard deviation of 4–7 mice per genotype. In (**B**, **C** and **E**) a *P*-value (Student's *t*-test) of $<0.01$ is indicated by **. Differences in (**C**) were not significant (*t*-test $P>0.05$).

As the role of PDK1 in B-cell development has not been established, the reason for the lack of mature B cells was investigated further. To determine if this was cell extrinsic or intrinsic, reconstitution experiments were carried out in sublethally irradiated Rag2 knockout mice. Injection of wild-type bone marrow that had been depleted of T and B cells into Rag2 mice reconstituted both T- and B-cell populations. In contrast, bone marrow from PDK1$^{fl/fl}$/Vav-Cre$^{+ve}$ mice was unable to effectively reconstitute either T or B cells in the Rag2 knockout mice (Supplementary Figure 7). To further examine this, we carried out competitive repopulation experiments. When a mixture of wild-type cells expressing either CD45.1 or CD45.2 markers were injected, both were able to give rise to mature B cells. However when a mixture of wild-type cells expressing CD45.1 and PDK1$^{fl/fl}$/Vav-Cre$^{+ve}$ expressing CD45.2 were injected, only the wild-type cells were found to give rise to B cells in either the blood or spleen (Figure 2).

### PDK1 is required for the transition of pro-B to pre-B cells

B cells develop in the bone marrow from CLPs (common lymphoid progenitors) and pass through a series of stages from pre-progenitor B (pre-pro-B) cells to progenitor B (pro-B) cells, precursor (pre-B) B cells and finally immature B cells which exit the bone marrow to mature in the periphery (Hardy and Hayakawa, 2001; Bartholdy and Matthias, 2004; Fuxa and Skok, 2007; Monroe and Dorshkind, 2007). As the PDK1$^{fl/fl}$/Vav-Cre$^{+ve}$ mice had no peripheral B cells, we examined B-cell development in the bone marrow. Total bone marrow cell numbers in the PDK1$^{fl/fl}$/Vav-Cre$^{+ve}$ mice were comparable to control genotypes (Figure 3A). FACS analysis for HSCs showed that they were present at the expected frequencies in the PDK1 knockout. CMP (common myeloid progenitors) and CLP populations were also present but there was an increased frequency of CMPs in PDK1$^{fl/fl}$/Vav-Cre$^{+ve}$ bone marrow relative to wild type (Figure 3B). CLPs may also be

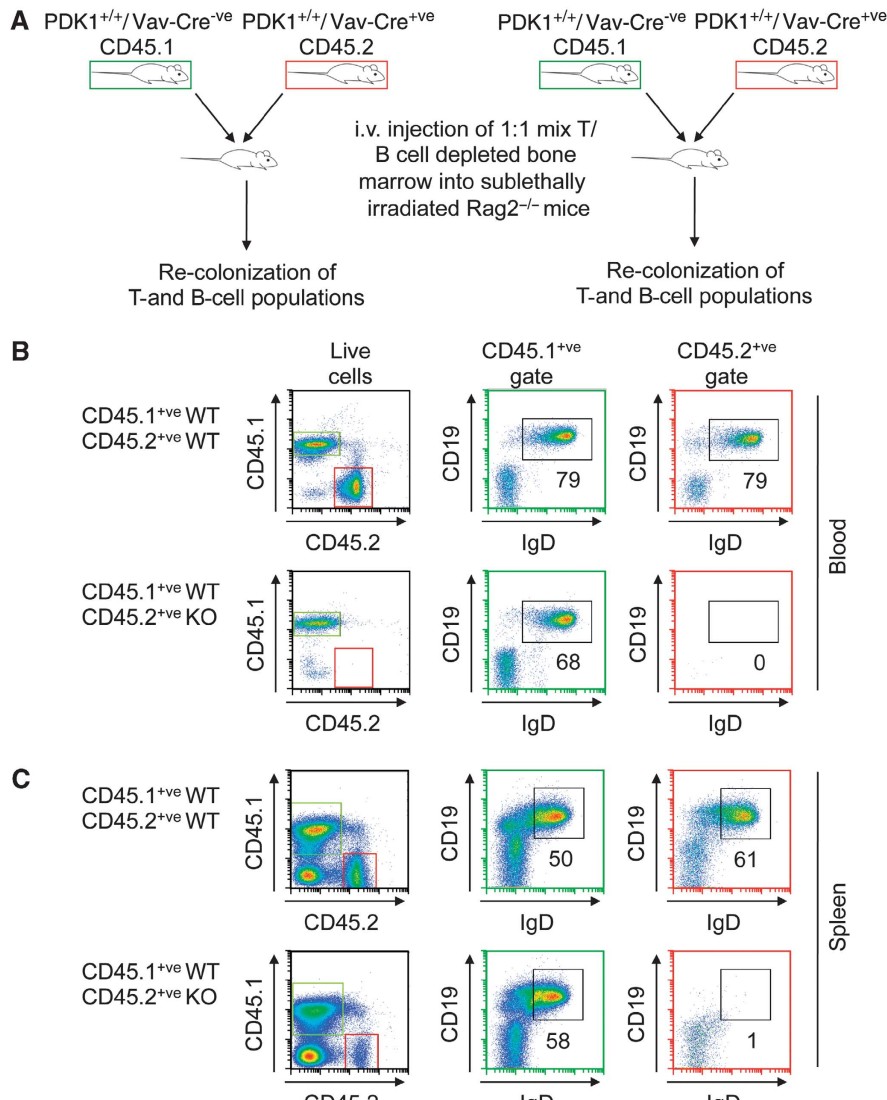

**Figure 2** The effect of PDK1 on B-cell development is cell intrinsic. Bone marrow was isolated from mice of the indicated genotypes and depleted of B220[+ve] and CD3[+ve] cells as described in Materials and methods. In all, 1:1 mixes of the depleted bone marrow from PDK1[+/+]/Vav-Cre[−ve] mice expressing the CD45.1 marker were mixed with either PDK1[+/+]/Vav-Cre[+ve] or PDK1[fl/fl]/Vav-Cre[+ve] cells expressing CD45.2. Cells were then injected into sublethally irradiated Rag2 knockout mice (**A**). The B-cell compartment in the blood (**B**) and spleen (**C**) is shown.

increased, although the resolution of this population in the knockouts was complicated by an increased c-kit[+ve]/Sca-1[low] cell population in this genotype. There was a trend for a lower number of total B-cell progenitors in the knockout bone marrow, however, this did not reach statistical significance ($P > 0.05$, Student's *t*-test). Normal numbers of pre-pro-B cells were present in PDK1[fl/fl]/Vav-Cre[+ve] mice; however, there was a greatly increased number of pro-B cells relative to control mice and significantly fewer pre-B cells. No immature or mature B cells were found in the bone marrow of PDK1[fl/fl]/Vav-Cre[+ve] mice (Figure 3C and D). An alternative staining protocol that also distinguishes pro-B (B220[+ve]IgM[−ve]c-Kit[+ve]CD25[−ve]) and pre-B cells (B220[+ve]IgM[−ve]c-kit[−ve]CD25[+ve]) (Xiao *et al*, 2007) also gave similar results (Supplementary Figure 8A). Loss of PDK1 therefore results in a block in B-cell development primarily at the pro- to pre-B cell transition. This effect was cell intrinsic as competitive repopulation experiments in

Rag2 knockout mice demonstrated that cells lacking PDK1 were unable to reconstitute the pre-B cell population in the bone marrow (Figure 3E).

### PDK1 regulates Ig heavy chain rearrangement

The transition from pro-B to pre-B cells requires the rearrangement of the IgH locus to allow expression of a pre-BCR (Hardy and Hayakawa, 2001; Bartholdy and Matthias, 2004). Analysis of the intracellular levels of IgH by FACS in pro-B cells demonstrated that fewer PDK1 knockout cells expressed high levels of heavy chain relative to control cells (Figure 4A). We therefore looked at the effect of PDK1 knockout on V-DJ recombination. V-DJ recombination was reduced in PDK1 knockout pro-B cells, an effect that was more apparent at distal ($V_H$J588) rather than proximal ($V_H$7183) $V_H$ regions (Figure 4B and C). Heavy chain recombination requires the expression of the recombinase genes *Rag1* and *Rag2*. Analysis of Rag1 and 2 mRNA expres-

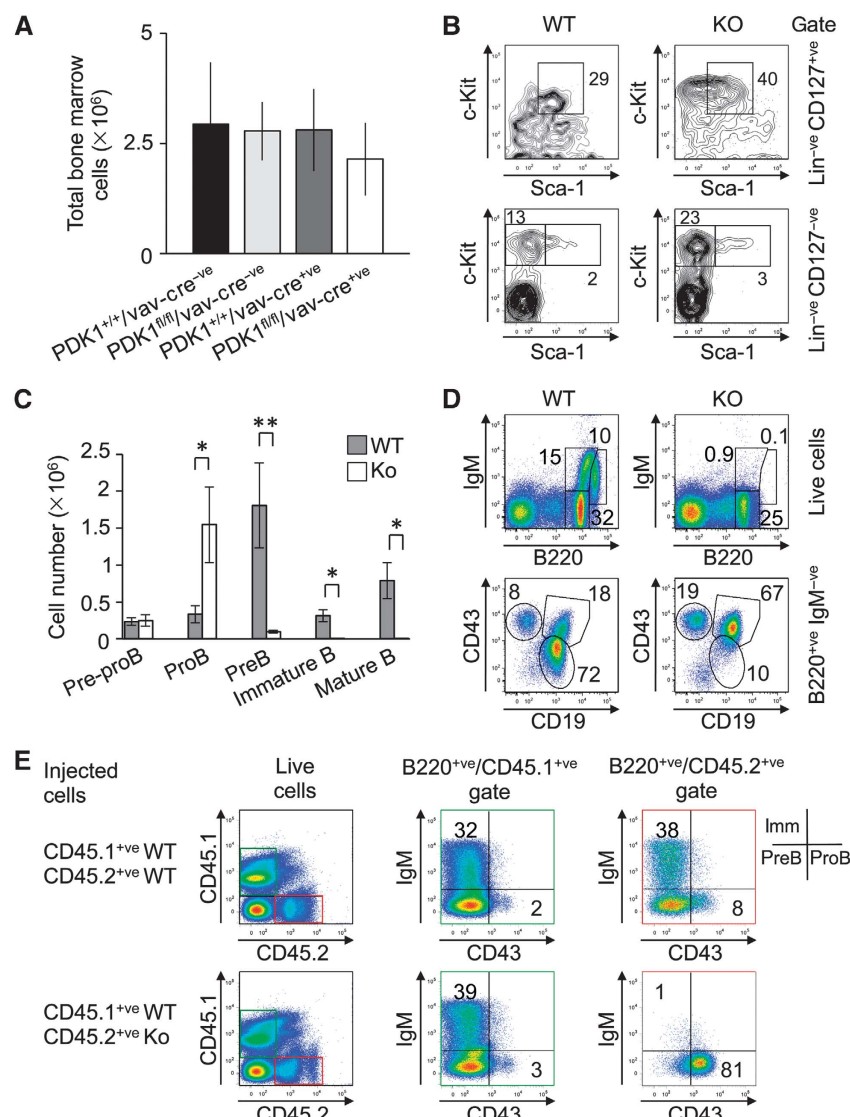

**Figure 3** Loss of PDK1 blocks B-cell development at the pro-B cell developmental stage. Total bone marrow cell numbers were similar between PDK1$^{fl/fl}$/Vav-Cre$^{+ve}$ and control genotypes (**A**). Error bars represent the standard deviation of 6–16 mice per genotype. Differences were not significant (*t*-test, $P > 0.05$). Cell development in the bone marrow was further examined by FACS. Staining of Lin$^{-ve}$CD127$^{+ve}$ cells with c-Kit and Sca1 was used to identify CLPs while staining of the Lin$^{-ve}$CD127$^{-ve}$ cells identified the MLP (c-kit$^{+ve}$/Sca1$^{-ve}$) and HSC (c-kit$^{+ve}$/Sca1$^{+ve}$) compartments. Results are representative of FACS on three mice per genotype (**B**). Staining of B220$^{+ve}$ cells for IgM demonstrated that loss of PDK1 blocked the development of immature (B220$^{low}$/IgM$^{+ve}$) and mature (B220$^{hi}$/IgM$^{+ve}$) cells (**C, D**). CD19/CD43 staining of the B220$^{+ve}$/IgM$^{-ve}$ cells showed that PDK1 knockout resulted in an accumulation of pro-B cells but a drastic decrease in pre-B cell numbers (**D**, lower panels). Representative plots are shown in (**D**) and quantification of absolute cell numbers based on this shown in (**C**). Error bars in (**C**) represent the standard deviation of 4–5 mice per genotype. A *P*-value (Student's *t*-test) of $<0.01$ is indicated by ** and 0.05 by *. Competitive repopulation experiments (**E**) into Rag2 knockout mice demonstrated that while injected wild-type cells could give rise to immature B cells, PDK1$^{fl/fl}$/Vav-Cre$^{+ve}$ cells were unable to differentiate into B220$^{+ve}$IgM$^{+ve}$/CD43$^{-ve}$ immature B cells but were instead arrested as pro-B cells (B220$^{+ve}$IgM$^{-ve}$/CD43$^{+ve}$).

sion showed that loss of PDK1 resulted in a two- to four-fold increase in the mRNA for these proteins (Figure 4D). Thus, failure to transcribe *Rag* genes would not account for the defect in V-DJ recombination. Although PDK1 knockout pro-B cells did not exhibit significantly different Pax5 mRNA ($P < 0.05$) to wild-type cells (Figure 4D), Pax5 protein expression was slightly decreased in these cells (see Figure 7D). The mRNA levels of Artemis, Ku70, Ku80, DNA-PKc and XRCC4, which are also involved in recombination (Helmink and Sleckman, 2012; Schatz and Ji, 2011), were not significantly altered by PDK1 knockout, while levels of TdT mRNA were increased. By contrast, PDK1

knockout resulted in an ~60% decrease in the expression of DNA Ligase IV (Figure 4D), an enzyme required for non-homologous end joining during IgH rearrangement (Frank *et al*, 1998; Grawunder *et al*, 1998).

### Loss of PDK1 inhibits Akt and RSK activation

While PDK1 deficiency reduced recombination at the IgH locus, it was not totally blocked and therefore may not completely explain the absence of B cells. Once heavy chain is expressed, it combines with the VpreB, λ5, Igα and Igβ chains to form a functional pre-BCR (Milne *et al*, 2004; Herzog *et al*, 2009). mRNA expression of these pre-BCR

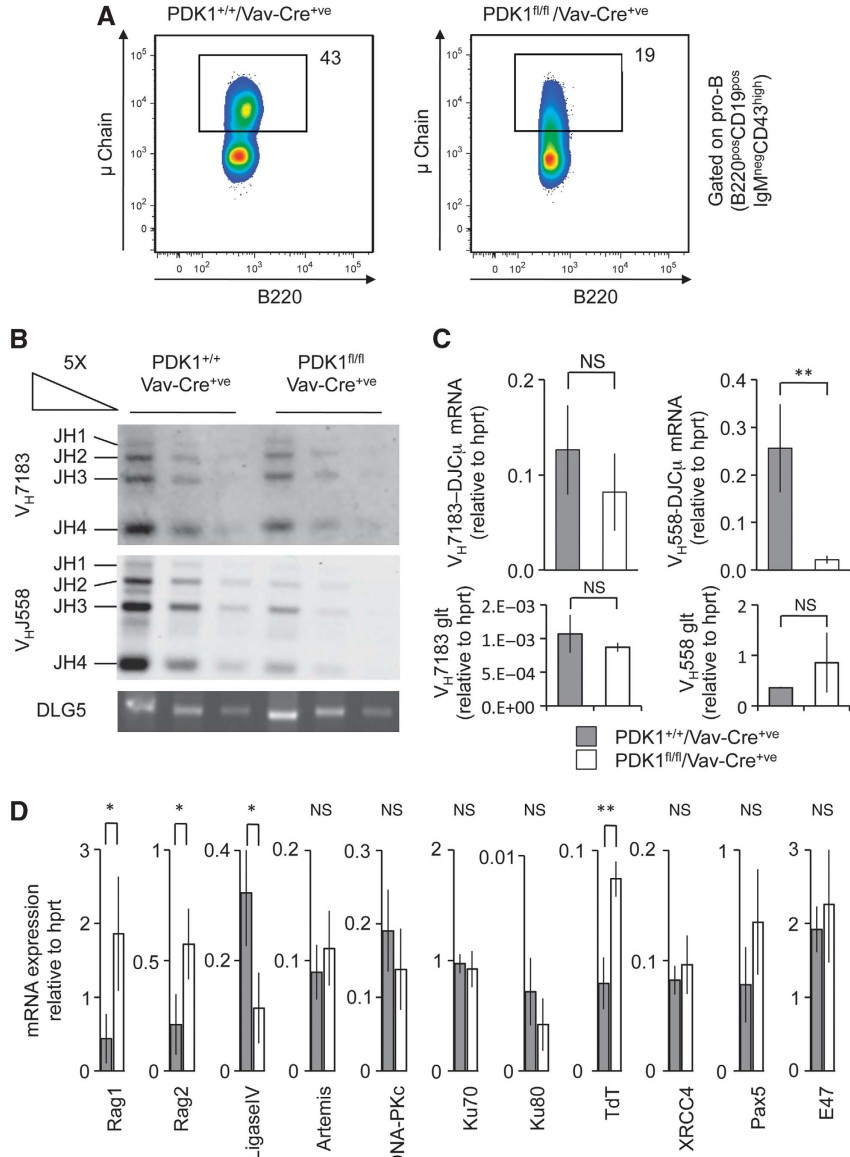

**Figure 4** PDK1 deficiency impairs V-DJ recombination. Staining for the presence of intracellular heavy chain in CD19$^{+ve}$IgM$^{-ve}$CD43$^{high}$B220$^{+ve}$ bone marrow (pro-B) cells demonstrated that while PDK1$^{+/+}$Vav-Cre$^{+ve}$ cells were able to express the heavy chain, the levels of intracellular heavy chain were reduced in the PDK1$^{fl/fl}$/Vav-Cre$^{+ve}$ cells. Results are representative of six mice (**A**). To examine V-DJ recombination, pro-B cells were isolated from PDK1$^{+/+}$/Vav-Cre$^{+ve}$ and PDK1$^{fl/fl}$/Vav-Cre$^{+ve}$ mice using FACS sorting. The heavy chain region was amplified by PCR and Southern blotting used to examine recombination for distal (V$_H$J588) and proximal (V$_H$7183) rearrangements (**B**). V-DJ mRNA expression as well as the expression of the V$_H$7183 and V$_H$J588 germline transcripts was also analysed using qPCR (**C**) with similar results. Total RNA was isolated from FACS-sorted pro-B cells and the levels of genes involved in V-DJ recombination were determined by qPCR (**D**). Expression levels were determined relative to HPRT and error bars represent the standard deviation of RNA preparations from FACS-sorted pro-B cells from three independent pools of mice per genotype. Loss of PDK1 resulted in significant increases in the mRNA levels for Rag1, Rag2 and TdT and decreased levels of DNA Ligase IV. In (**C**, **D**), a $P < 0.05$ is indicated by * and $P < 0.01$ by ** while NS indicates $P > 0.05$ (Student's *t*-test).

components in the PDK1$^{fl/fl}$/Vav-Cre$^{+ve}$ pro-B cells was similar to wild-type (Figure 5A). Analysis of the levels of intracellular heavy chain in the PDK1 knockout pre-B cell gate demonstrated that it was expressed in the majority of these cells (Figure 5B; Supplementary Figure 8). The increased expression of the intracellular heavy chain in the pre-B cell gate relative to pro-B cells (compare Figures 4A and 5B) is consistent with some PDK1 knockout cells being able to make the transition from pro-B to pre-B cells. The low numbers of these cells and their failure to develop further would however suggest that PDK1 plays important roles at the pre-B cell stage.

In these cells, PDK1 could act downstream of several signals, including IL-7 and the pre-BCR. To confirm that the knockout of PDK1 in developing B cells was sufficient to block the activation of its downstream targets, pro-B cells were expanded in IL-7 and then after IL-7 withdrawal stimulated with anti-Igβ and the activation of intracellular signalling pathways analysed by immunoblotting (Figure 5C). Both wild-type and PDK1$^{fl/fl}$/Vav-Cre$^{+ve}$ cells were able to activate ERK1/2, indicating that both genotypes could respond to the anti-Igβ stimulation. RSK is activated by a complex mechanism involving phosphorylation by both ERK1/2 and PDK1

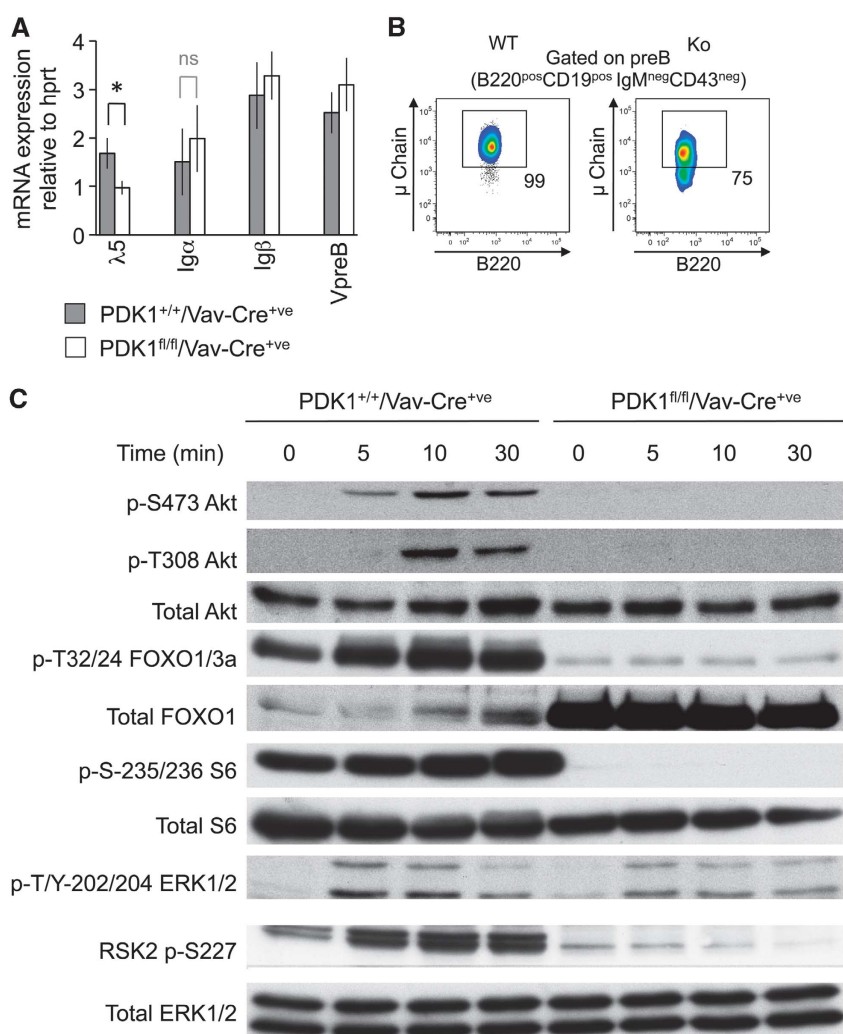

**Figure 5** PDK1 is required for Akt, p70S6K and RSK2 activation in B cells. (**A**) The mRNA levels for the pre-BCR components λ5, Igα, Igβ and VpreB were determined by qPCR from total RNA isolated from FACS-sorted pro-B cells from either PDK1$^{+/+}$Vav-Cre$^{+ve}$ or PDK1$^{fl/fl}$Vav-Cre$^{+ve}$ mice. Error bars represent the standard deviation of RNA from three independent pools of mice per genotype. (**B**) The levels of intracellular heavy chain were determined by FACS staining of *ex vivo* bone marrow cells, gating on the CD19$^{+ve}$IgM$^{-ve}$CD43$^{-ve}$B220$^{+ve}$ cells to examine pre-B cells. (**C**) Pro-B cells were isolated from PDK1$^{+/+}$Vav-Cre$^{+ve}$ and PDK1$^{fl/fl}$Vav-Cre$^{+ve}$ mice and stimulated for 6 days with 10 ng/ml IL-7. After withdrawal of IL-7 for 4 h, cells were re-stimulated for the indicated times with 30 μg/ml anti Igβ and the levels of the indicated proteins determined by immunoblotting.

(Pearce *et al*, 2010). While ERK1/2 was activated normally in the PDK1$^{fl/fl}$/Vav-Cre$^{+ve}$ cells the phosphorylation of RSK on the PDK1 site, S227, was inhibited. As S227 is critical for RSK activity (Pearce *et al*, 2010), this indicates that RSK would be inactive in these cells. PDK1 is also required for the activation of p70S6K (Pearce *et al*, 2010), and consistent with this the PDK1$^{fl/fl}$/Vav-Cre$^{+ve}$ cells had no phosphorylation of the p70S6K substrate S6 (Figure 5C).

In wild-type cells, anti-Igβ activated Akt, as judged by phosphorylation of the PDK1 site T308 as well as the mTORC2 site S473. PDK1 knockout cells showed no phosphorylation of T308. Interestingly, S473 was also lost in the PDK1 pro-B cells, suggesting that the phosphorylation of T308 and S473 was interdependent in these cells. Akt phosphorylates FOXO1 and 3a, and in line with the loss of Akt activation, both basal and anti-Igβ activated FOXO1/3a phosphorylation were greatly reduced by the knockout of PDK1. In contrast, total FOXO1 levels were increased, which is consistent with previous observations that demonstrate blocking

Akt signalling inhibits FOXO1 phosphorylation but increases FOXO1 protein levels (Lazorchak *et al*, 2010) and is in line with a role for the phosphorylation of FOXO in regulating its ubiquitination and degradation (Huang and Tindall, 2011). The loss of basal FOXO phosphorylation suggests that PDK1 is also required for the activation of Akt by other stimuli in pro-B cells. In agreement with this, treatment of the 70z/3 pre-B cell line with a recently described inhibitor of PDK1 was sufficient to block IL-7 induced Akt and FOXO phosphorylation (Supplementary Figure 9). The ability to regulate IL-7 induced Akt activation is likely to be especially significant given the recent findings that IL-7, and not the pre-BCR is the major stimuli responsible for Akt activation in B cell development (Ochiai *et al*, 2012).

### PDK1 regulates the cell cycle and survival during B-cell development

Following V-DJ recombination, pro-B cells proliferate in response to IL-7 to differentiate into large cycling pre-B

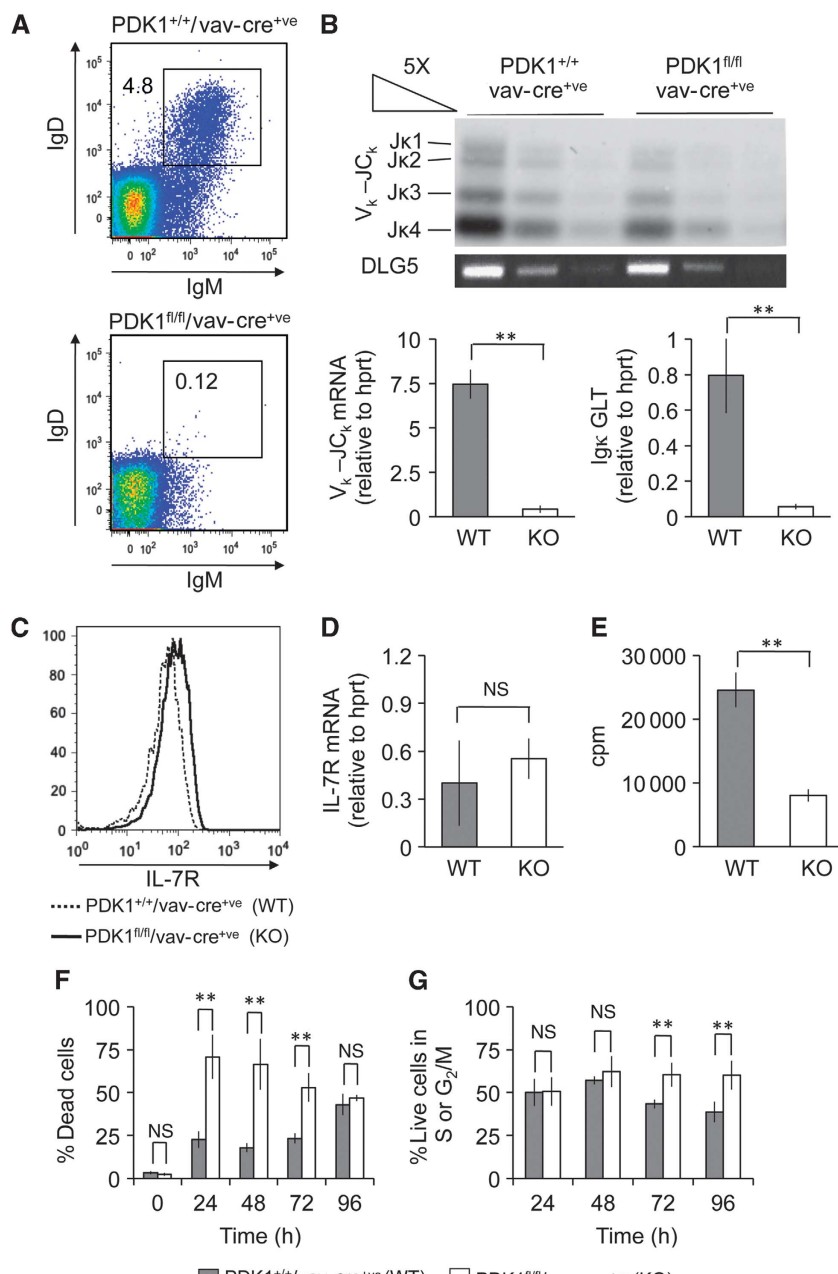

**Figure 6** PDK1 regulates proliferation and survival of pro and pre-B cells. To analyse the *in vitro* differentiation, pro-B cells were stained for surface IgM and IgD after they were grown in 10 ng/ml IL-7 for 4 days followed by 48 h of IL-7 withdrawal (**A**). To determine the level of Igκ (Light) chain recombination *in vivo*, pre-B cells were isolated from PDK1$^{+/+}$/Vav-Cre$^{+ve}$ and PDK1$^{fl/fl}$/Vav-Cre$^{+ve}$ mice by FACS sorting and V$_\kappa$-J$_\kappa$ recombination analysed by Southern blotting as described in Materials and methods (**B**, upper panel). The levels of mRNA for Igκ rearrangement as well as Igκ germline transcript (GLT) were determined by qPCR (**B**, lower panels). Error bars represent the standard deviation of three independent samples per genotype. The levels of IL-7 receptor protein and mRNA in PDK1$^{+/+}$Vav-Cre$^{+ve}$ and PDK1$^{fl/fl}$Vav-Cre$^{+ve}$ pro-B cells were determined by FACS (**C**) and qPCR (**D**), respectively. Error bars represent the standard deviation of RNA from FACS-sorted pro-B cells from three independent pools of mice per genotype. To measure *in vitro* responses to IL-7 (**E–G**), pro-B cells were isolated from PDK1$^{+/+}$Vav-Cre$^{+ve}$ and PDK1$^{fl/fl}$Vav-Cre$^{+ve}$ mice and cultured in the presence of 10 ng/ml IL-7 for 48 h. Overall proliferation over 66 h was measured by the uptake of $^3$H-Thymidine (**E**). Cell death was determined by uptake of propidium iodide (PI) into unpermeabilized cells (**F**). The percentage of live cells (based on forward and side scatter) in the S and G2/M phases of the cell cycle was determined by FACS of intracellular PI-stained cells (**G**). For (**F, G**), error bars represent the standard deviation of three independent preparations of cells per genotype. A *P*-value (Student's *t*-test) of <0.01 is indicated by ** while NS indicates *P*>0.05.

cells which then exit the cell cycle and undergo light chain recombination at the small pre-B cell stage (Milne *et al*, 2004). pro-B cells can be induced to proliferate *in vitro* by the addition of IL-7, and will undergo light chain recombination, an event that can be further promoted by withdrawal of IL-7. In line with this, cultured wild-type cells

were able to express surface IgM, an event that was promoted by IL-7 withdrawal (Aiba *et al*, 2008). In contrast, PDK1$^{fl/fl}$/Vav-Cre$^{+ve}$ cells failed to upregulate IgM/IgD expression (Figure 6A). Consistent with this, PDK1$^{fl/fl}$/Vav-Cre$^{+ve}$ cells also showed reduced light chain recombination *in vivo* relative to PDK1$^{+/+}$/Vav-Cre$^{+ve}$ controls (Figure 6B).

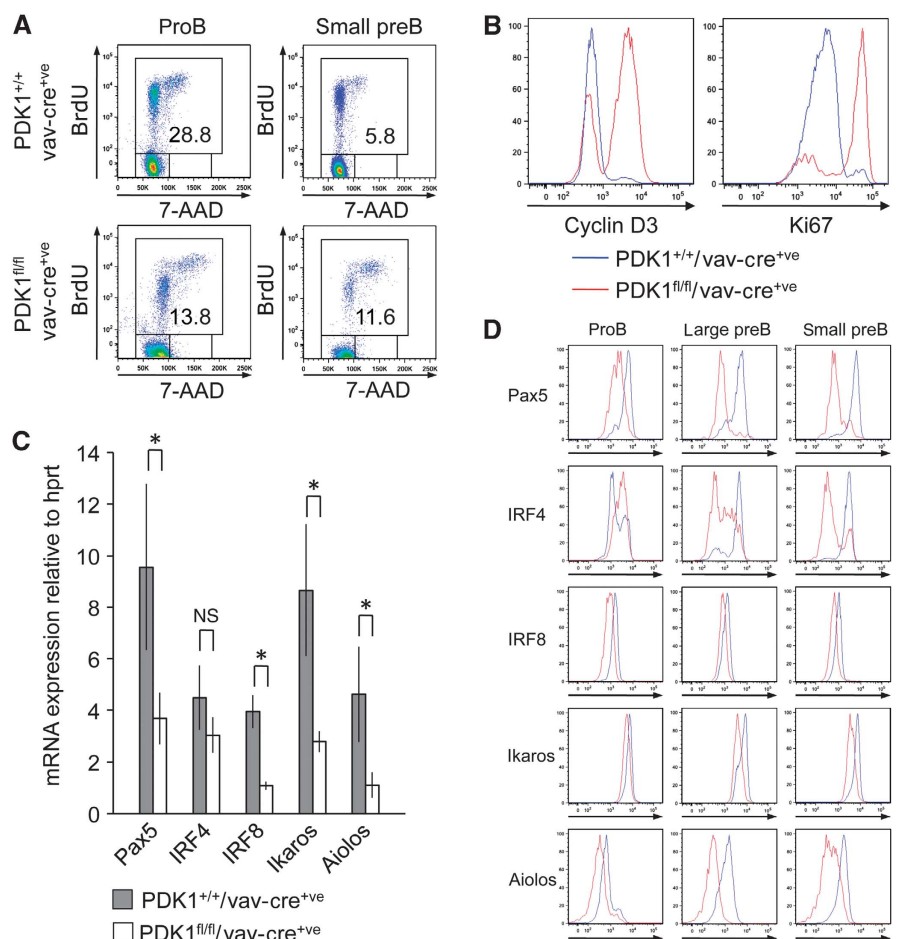

**Figure 7** PDK1 regulates Pax5 expression and cell-cycle progression in pre-B cells. To look at proliferation *in vivo*, PDK1$^{+/+}$/Vav-Cre$^{+ve}$ and PDK1$^{fl/fl}$/Vav-Cre$^{+ve}$ mice were injected with 1 mg BrdU 3.25 h before being sacrificed. Bone marrow was then stained for 7-AAD and pre-B cell markers as described in Materials and methods. BrdU and 7-AAD uptake was determined by FACS in the pro-B and small pre-B cells (**A**). The levels of the proliferation marker Ki67 as well as cyclinD3 in *ex vivo* pre-B cells were also determined by FACS (**B**). To examine the expression of Pax5, IRF4, IRF8, Ikaros and Aiolos, total RNA was isolated from FACS-sorted pre-B cells and the mRNA levels for these genes determined by qPCR (**C**). Error bars represent the standard deviation of RNA from FACS-sorted pre-B cells of three independent pools of mice per genotype, $P < 0.05$ (Student's test) is indicated by * while NS indicates $P > 0.05$. The levels of Pax5, IRF4, IRF8, Ikaros and Aiolos protein (**D**) were determined by intracellular FACS staining of pro-B (CD19$^{+ve}$IgM$^{-ve}$CD43$^{high}$B220$^{+ve}$), large pre-B (FSC$^{high}$CD19$^{+ve}$IgM$^{-ve}$CD43$^{-ve}$B220$^{+ve}$) or small pre-B (FSC$^{low}$CD19$^{+ve}$IgM$^{-ve}$CD43$^{-ve}$B220$^{+ve}$). Results are representative of three mice per genotype.

This could suggest a failure to express the IL-7 receptor in the PDK1 knockout cells. PDK1 knockout pro-B cells however were found to express the CD127, IL-7 receptor subunit at slightly higher levels than wild-type cells as determined by qPCR or FACS (Figure 6C,D). PDK1$^{fl/fl}$/Vav-Cre$^{+ve}$ cultures showed greatly reduced levels of *in vitro* $^3$H thymidine uptake in response to IL-7 relative to wild-type cells, suggesting a role for PDK1 in proliferation or survival (Figure 6E). Propidium iodide staining revealed that during the first 3 days of culture, the rate of cell death was increased in the PDK1$^{fl/fl}$/Vav-Cre$^{+ve}$ cells relative to control cultures (Figure 6F). Cell-cycle analysis demonstrated that by day 3 PDK1$^{fl/fl}$/Vav-Cre$^{+ve}$ cultures had an increase in the number of cells in S-G2/M phase of the cell cycle relative to wild-type cultures, which is consistent with the failure of the PDK1 knockout cells to become pre-B cells and initiate light chain recombination (Figure 6G).

Together, these experiments demonstrate that PDK1 is required for survival in the *in vitro* IL-7 culture model as well as for light chain recombination. This would suggest that

loss of PDK1 should result in increased cell death in developing B cells *in vivo*, together with a failure of large pre-B cells to exit the cell cycle and initiate light chain recombination. To examine this *in vivo*, mice were injected with BrdU and incorporation of it was determined in the pro and pre-B cell subsets (Figure 7A). PDK1 knockout resulted in a decrease in the number of pro-B cells taking up BrdU, which is in line with the reduced heavy chain recombination seen in the PDK1 knockout pro-B cells (Figure 4). Once heavy chain recombination is complete, pro-B cells start to proliferate and differentiate becoming large pre-B cells that subsequently exit from the G2/M phase of the cell cycle to become small pre-B cells and initiate light chain recombination (Kurosaki *et al*, 2010). Loss of PDK1 resulted in an increased percentage of BrdU-labelled small pre-B cells relative to wild-type controls, suggesting that the loss of PDK1 reduces the ability of the cells to exit the S/G2/M phase of the cell cycle in order to initiate light chain recombination (Figure 7A). In addition, a higher percentage of cells expressing Cyclin D3, a protein that promotes S-phase entry, and Ki67, a marker for proliferation

were observed in the PDK1 knockout pre-B cells (Figure 7B). This is in line with the in vitro cell-cycle results (Figure 6G). Taken together, this demonstrates that PDK1 promotes the transition from large pre-B to small pre-B cells.

The transition from large to small pre-B cells is accompanied by downregulation of the λ5 and VpreB components of the pre-BCR due to increased levels of Ikaros and Aiolos, which are induced by IRF4 and IRF8 (Ma et al, 2008). Analysis of the expression of these genes in the PDK1 null pre-B cells showed that in comparison to wild-type cells, loss of PDK1 resulted in higher expression of λ5 and VpreB (Supplementary Figure 10) but decreased expression of IRF4, IRF8, Ikaros and Aiolos (Figure 7C). The expression of IRF4, IRF8 and Aiolos is dependent on Pax5 in pre-B cells (Schebesta et al, 2007). Pax5 expression was found to be reduced at both the mRNA and protein levels at the pre-B cell stage in the PDK1 knockout cells relative to wild-type controls (Figure 7C and D), providing a potential explanation for the reduced expression of Aiolos and IRF8. In line with this two further Pax5 targets, the Igκ germline transcript (Figure 6B) and Blnk (Supplementary Figure 11) were decreased in PDK1 null pre-B cells. The reduction in IRF4 protein expression was not apparent at the mRNA level, suggesting that PDK1 may be affecting the translation or turnover of the IRF4 protein. Aiolos represses Ccnd3 expression in pre-B cells (Mandal et al, 2009) and in agreement with this, the decreased expression of Aiolos in PDK1 null pre-B cells correlated with an increased expression of Ccnd3 (Figure 7B–D), which may explain their inability to exit cell cycle in order to undergo Ig light chain rearrangement.

### Bcl2A1 and Pax5 restore the differentiation of B cells in the absence of PDK1

In vitro, PDK1$^{fl/fl}$/Vav-Cre$^{+ve}$ pro-B cells also showed a significantly increased rate of cell death (Figure 6F) in response to IL-7. A similar increase in apoptosis was seen in vivo in pre-B but not in pro-B cells, as judged by either Annexin V (Figure 8A) or phospho-histone H2AX staining in pre-B cells (Supplementary Figure 11). The difference between the effect of PDK1 knockout on pro-B cell survival in vivo compared to the in vitro IL-7 stimulated cultures most likely reflects the inability of the culture system to provide all the necessary trophic support to the pro-B cells. The pro-survival genes Mcl-1 and Bcl2A1 have both been shown to affect survival in early B-cell development (Chuang et al, 2002; Opferman et al, 2003). In pre-B cells, knockout of PDK1 resulted in a large decrease in Bcl2A1 mRNA expression as well as a more moderate decrease in Mcl-1 relative to control genotypes (Figure 8B). Of note, Bcl2A1 is an NF-κB target (Vogler, 2012) and the expression of another classical NF-κB target, IκBα, was also reduced in the PDK1 knockout pre-B cells (Supplementary Figure 11). In vitro, reintroduction of Bcl2A1 into PDK1 knockout pro-B cells using retroviral transduction was sufficient to increase the survival of these cells as shown by forward and side scatter plots (Figure 8C and D) or by DAPI staining (Supplementary Figure 12A); however, Bcl2A1 expression did not restore that ability of the cells to upregulate the expression of IgM and IgD (Figure 8C and E). In line with this, Bcl2A1 expression did not affect the ability of the cells to downregulate CD43 or to decrease the percentage of cells in the G2/M phase of the cell cycle (Supplementary Figure 12B).

Transfection of Pax5 into wild-type cells reduced cell viability, however in line with its role in promoting light chain rearrangement, it did increase the number of IgM/IgD double-positive cells relative to untransfected controls (Supplementary Figure 12C). Re-expression of Pax5 alone in pro-B cells however was insufficient to rescue either the survival of the PDK1 knockout pre-B cells or their ability to differentiate into IgM/IgD double-positive cells (Figure 8C and E). We therefore transduced PDK1 knockout pro-B cells with a combination of retroviruses encoding both GFP-tagged Pax5 and mCherry-tagged Bcl2A1. Gating on the GFP mCherry double-positive cells demonstrated an increased percentage of cells expressing IgM/IgD relative to cells transfected with mCherry-Bcl2A1, GFP-Pax5 or GFP alone (Figure 8C and E). A similar result was also obtained for IgM-positive cells (Supplementary Figure 12D). In line with this, a combination of Bcl2A1 and Pax5 transfection was able to promote the downregulation of CD43 (Supplementary Figure 12B) and reduce the number of cells in the G2/M phase of the cell cycle (Figure 8F; Supplementary Figure 12E). The above results indicate that PDK1 is required to control B-cell development and that this occurs at least in part by the regulation Pax5 and Bcl2A1 expression.

## Discussion

We show here that while PDK1 is dispensable for myeloid cell development, it is critical for both B- and T-cell differentiation. Interestingly the role of PDK1 varies between T and B cells; while PDK1 is dispensable for recombination of the TCRβ locus (Supplementary Figure 6; Hinton et al, 2004), it does regulate Ig locus rearrangement in B cells. PDK1 knockout results in an arrest of B-cell development at the transition from the pro-B to pre-B cell stage. This corresponded to a decrease, but not total block, of IgH recombination and an increase in apoptosis in pre-B cells along with a decrease in the ability to initiate light chain recombination (Figures 4 and 6).

PDK1 is required for the activation of several AGC kinases including Akt in the PI-3 kinase cascade (Pearce et al, 2010). Several recent studies support a role for PI-3 kinase and Akt signalling in B-cell development (Lazorchak et al, 2010; Ramadani et al, 2010; Baracho et al, 2011). Mice with a conditional deletion of PI3Kα and an inactivating mutation in PI3Kδ also exhibit a block in B-cell development at the pro- to pre-B stage (Ramadani et al, 2010). This was attributed to an increase in Rag expression and enhanced IgH recombination as well as decreased proliferation in response to IL-7, although it was not established if this was due to decreased cell division or increased apoptosis. In agreement with this, mice deficient in Sin1, a component of the mTORC2 complex, also show a partial bock in B-cell development with enhanced Rag expression and augmented IgH (V-DJ) recombination (Lazorchak et al, 2010). This is consistent with other studies showing that FOXO1 is required for the expression of the Rag gene in B cells (Amin and Schlissel, 2008; Dengler et al, 2008; Herzog et al, 2008). As Akt phosphorylates and inhibits FOXO1, blockade of the PI3K/Akt pathway would increase FOXO activity. Our results using PDK1 knockouts also demonstrate a role for this pathway in the regulation of FOXO1 and Rag gene expression (Figure 4D). However, the role of PDK1 in reg-

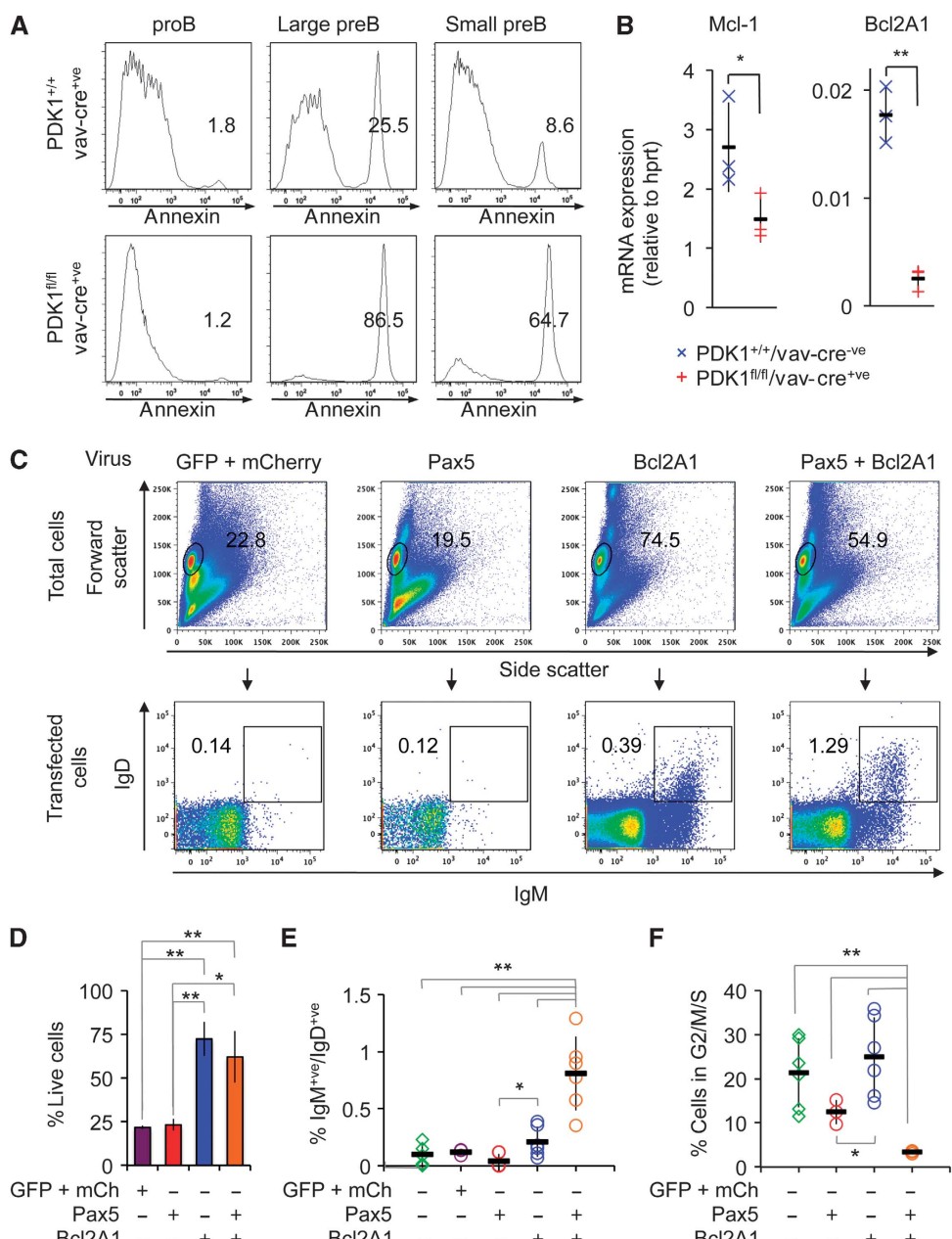

**Figure 8** Pax5 and Bcl2A1 induce B-cell differentiation in the absence of PDK1. The levels of apoptosis in *ex vivo* PDK1$^{+/+}$Vav-Cre$^{+ve}$ and PDK1$^{fl/fl}$Vav-Cre$^{+ve}$ cells were determined by annexin V staining of pro-B (CD19$^{+ve}$IgM$^{-ve}$CD43$^{high}$B220$^{+ve}$), large pre-B (FSC$^{high}$CD19$^{+ve}$IgM$^{-ve}$CD43$^{-ve}$B220$^{+ve}$) or small pre-B (FSC$^{low}$CD19$^{+ve}$IgM$^{-ve}$CD43$^{-ve}$B220$^{+ve}$) cells. Results are representative of three mice (**A**). qPCR was used to determine the mRNA levels of Mcl-1 and Bcl2A1 in FACS-sorted pre-B cells (*n* = 3) from PDK1$^{+/+}$Vav-Cre$^{+ve}$ and PDK1$^{fl/fl}$Vav-Cre$^{+ve}$ mice (**B**). To determine if Pax5 and Bcl2A1 were involved in mediating PDK1 function in B cells, PDK1$^{fl/fl}$Vav-Cre$^{+ve}$ FACS-sorted CD19$^{+ve}$IgM$^{-ve}$CD43$^{high}$B220$^{+ve}$DAPI$^{-ve}$ pro-B cells were infected with the indicated combinations of retroviruses for GFP, mCherry, GFP-Pax5 or mCherry-Bcl2A1 as described in Materials and methods. The percentages of live cells on day 6 in culture were determined based on forward and side scatter (**C**, upper panels). BCR expression was then determined by surface staining for IgM and IgD in the GFP and/or mCherry-positive cells (**C**, lower panels). Quantification of the % of live cells in (**C**) (*n* = 3) is shown in (**D**) while qualification of the % IgM/IgD positive gates for the indicated proteins in shown in (**E**). To analyse the effect on the cell cycle, PDK1 knockout pro-B cells were isolated and transfected with retroviruses for GFP-Pax5 and mCherry-Bcl2A1. After 6 days, the cell cycle was analysed by Hoechst staining. B220 + ve cells were gated for the expression of Blc2A1 and Pax5 and the % of cells in the G2/M/S phase of the cell cycle determined (**F**). Error bars represent the standard deviation and a *P*-value (Student's *t*-test) of <0.01 is indicated by ** and <0.05 by *.

ulating Akt downstream of PI-3 kinase does not appear to be sufficient to explain the phenotype of the PDK1 knockout in B cells. Notably, while PI3Kα/PI3Kδ mutant mice and also Sin1 knockout mice show increased V-DJ recombination (Lazorchak *et al*, 2010; Ramadani *et al*, 2010), knockout of PDK1 reduces V-DJ recombination, especially at distal loci

(Figure 4B and C), which correlated with decreased expression of Pax5 and DNA ligase IV. DNA ligase IV is involved in non-homologous end joining of dsDNA breaks, and mutations in this gene in humans or mice result in defects in Ig locus recombination (Frank *et al*, 1998; Grawunder *et al*, 1998; Chistiakov, 2010). This variation in phenotype between

the different mouse models indicates that either PDK1 has PI-3 kinase/Akt-independent roles, or that the PI3Kα/PI3Kδ pro-B cells still had a low level of Akt activity. The first explanation seems more likely for several reasons. First, a reduction in PDK1 levels (Hinton *et al*, 2004) or mutations that reduce its ability to activate Akt (data not shown) do not block B-cell development, this would be at odds with the idea that a low level of Akt activity in the PI3Kα/PI3Kδ mutant mice explains the difference in phenotype with the PDK1 knockout. Secondly, a recent study has shown that in Irf4$^{-/-}$/Irf8$^{-/-}$ cycling pre-B cells, Akt activation is regulated predominately via IL-7 and not the pre-BCR. In these cells, attenuation of IL-7 was found to reduce Akt phosphorylation and upregulate the expression of FOXO1 and 3a. Interestingly, Pax5 levels were also upregulated under these conditions, and the effect of IL-7 attenuation could be mimicked using the PI3-kinase inhibitor LY294002 (Ochiai *et al*, 2012). The knockout of PDK1 has similar effects on the levels of FOXO1 (Figure 5); however, loss of PDK1 decreases Pax5 levels (Figure 7). Finally, it is well established that PDK1 can regulate other AGC kinases, including RSK and PKCs, independently of PI-3 kinase signalling (Pearce *et al*, 2010). Thus, PDK1 has the potential to influence multiple signalling pathways, and as a result its effects are likely to be complex.

PDK1 was also found to regulate the survival of pre-B cells and was required to promote the onset of Igκ chain rearrangement. As mentioned above, levels of Pax5 mRNA and protein expression were reduced in PDK1 null pre-B cells. Pax5 promotes the induction of several genes including *IRF4* and *IRF8* (Schebesta *et al*, 2007; Pridans *et al*, 2008). Double knockout of IRF4 and 8 arrests B cells at the cycling large pre-B stage and prevents light chain recombination (Lu *et al*, 2003). This was linked to the ability of IRF4 and 8 to upregulate the expression of Aiolos and Ikaros, which in turn promote downregulation of the pre-BCR and inhibit the G1/S cell-cycle transition (Mao *et al*, 2007; Johnson *et al*, 2008; Ma *et al*, 2008). We show here that consistent with the reduced expression of Pax5 in the PDK1 knockouts, the expression of IRF4, Ikaros and Aiolos was also reduced. This corresponded to a reduced ability of the PDK1 knockout cells to stop proliferation and initiate light chain recombination. The regulation of *Pax5* is complex, and is controlled by both E2A and EBF1 binding to the promoter and an enhancer that binds PU.1, IRF4, IRF8 and NF-κB (Decker *et al*, 2009). Thus, PDK1 could reinforce Pax5 expression more strongly in pre-B cells than in pro-B cells via several mechanisms, including through the IRF4/8 feedback loop on the Pax5 promoter or via the NF-κB enhancer. PDK1 has been implicated in NF-κB regulation in T cells via its ability to activate PKCθ (Lee *et al*, 2005), although recently the GLK, rather than PDK1, has been suggested to be responsible for PKCθ activation (Chuang *et al*, 2011). PKCβ has been proposed to play a similar role in mature B cells; however, it is not clear what the relevant PKC isoform would be in pre-BCR signalling (Guo *et al*, 2004). Interestingly, however, in pre-B cells we found that the expression of several NF-κB targets, such as Bcl2A1 and IκBα was decreased by PDK1 knockout. Due to the complexity of the NF-κB system, its precise role in B-cell development has been hard to determine (Sasaki *et al*, 2007). NF-κB activity has been reported to increase as cells transition from pro- to pre-B cells (Jimi *et al*, 2005); however, knockouts of NF-κB subunits do not block early B-cell development although double knockout of both NF-κB1 and NF-κB2 was found to impede the development of immature B cells (Claudio *et al*, 2009). It is likely however that the effect of inhibiting NF-κB would be exacerbated in cells where FOXO signalling is already deregulated, as would be the case in the PDK1 knockouts in which the regulation of FOXO by Akt is abolished. An alternative explanation could be the reduced expression or activity of the preBCR in the PDK1 knockout pre-B cells. This decreased level of preBCR signalling could result in a failure to maintain Pax5 expression. It is also possible that the PDK1/Akt pathway could also affect Pax5 expression at post-transcriptional level, either via the regulation of protein translation or protein stability. Of note, multiple studies have linked the Akt pathway to the control of translation (reviewed in Ruggero and Sonenberg, 2005 and Venigalla and Turner, 2012).

Regardless of the mechanism by which PDK1 affects Pax5 levels in pre-B cells, it does play an important role in mediating the PDK1 phenotype in B-cell development. Reintroduction of Pax5 alone was insufficient to rescue the PDK1 phenotype in early B cells as it did not prevent the cells undergoing apoptosis, demonstrating a role for PDK1 in survival that is independent of Pax5 (Figure 8). In pre-B cells, we found that PDK1 regulated the expression of the pro-survival gene Bcl2A1. In mice, there are four closely related copies of the Bcl2A1 gene, making genetic analysis of the role of Bcl2A1 in B-cell development problematic (Mandal *et al*, 2005). Transgenic mice overexpressing Bcl2A1 however have shown that this protein can promote B-cell survival (Chuang *et al*, 2002), while other studies have shown a role for Bcl2A1 in promoting survival downstream of the pre-TCR (Mandal *et al*, 2005). *In vitro*, we found that Bcl2A1 expression was sufficient to rescue the apoptotic phenotype in the PDK1 knockout pre-B cells, but the development of IgM/IgD-positive cells was still retarded. A combination of Bcl2A1 and Pax5 transduction had an additive effect on the *in vitro* differentiation of PDK1 knockout pro-B cells (Figure 8). This is supportive of the idea that PDK1 is regulating early B-cell development via influencing both Ig locus recombination and survival.

Finally with regard to the role of PDK1 *in vivo*, in T cells PDK1 has been shown to regulate the expression of chemokines and adhesion receptors (Finlay *et al*, 2009). Although not addressed in our study it is possible that in the developing B cells in the bone marrow PDK1 is also required for the correct localization of the progenitor B cells via regulating either chemokines or adhesion receptors.

In summary, we show that PDK1 is an important regulator of B-cell differentiation in the bone marrow. The molecular mechanism behind this is complex due to the multiple pathways that PDK1 impacts on. In B cells, we provide evidence that PDK1 regulates both Rag gene expression via Akt/FOXO signalling pathway and the expression of Pax5 and Bcl2A1 in pre-B cells, most probably via regulation of NF-κB activation.

## Materials and methods

### Reagents
Recombinant murine IL-7 was obtained from R&D systems; SCF-1 and IL-6 were from Peprotech. The antibodies used are listed in Supplementary Table 1.

     

### Mice

Mice with a floxed allele of PDK1 and Vav-Cre transgenic mice have been described previously (Lawlor et al, 2002; deBoer et al, 2003). Animals were housed under specific pathogen-free conditions in individually ventilated cages in accordance with UK and EU regulations. All work was carried out under a UK project License and subject to local ethical review. Mice between 4 and 6 weeks of age were used for the experiments.

### Adoptive transfer

Total bone marrow cells were depleted of B220 and CD3-positive cells by using anti-biotin microbeads on MACS. Depleted bone marrow cells from either control or PDK1$^{fl/fl}$/Vav-Cre$^{+ve}$ donor mice were then transferred by intravenous injection into sublethally irradiated (6 Gray) Rag2$^{-/-}$ host mice. For competitive repopulation assays, B220 and CD3-depleted bone marrow cells from donor CD45.1 control mice were mixed with the same type of cells derived from either CD45.2 control or CD45.2 PDK1$^{fl/fl}$/Vav-Cre$^{+ve}$ donor mice in 1:1 ratio. After 2–9 weeks, recipient mice were analysed for reconstitution of bone marrow or spleen.

### Isolation of B-cell progenitors and cell culture

Single cell bone marrow suspensions were treated with red blood cell lysis buffer (Sigma) and then Gr1, CD11b, Ter119 and CD3-positive cells were depleted using MACS columns. Cells were stained with anti-CD19-APC, anti-IgM-FITC, anti-CD43-PE, anti-B220-PE-Cy5 and DAPI and sorted for CD19$^{-ve}$B220$^{+ve}$IgM$^{-ve}$CD43$^{high}$DAPI$^{-ve}$ (pre-progenitor B cell), CD19$^{+ve}$B220$^{+ve}$IgM$^{-ve}$CD43$^{high}$DAPI- (progenitor B cell), CD19$^{+ve}$B220$^{+ve}$IgM$^{-ve}$CD43$^{-ve}$DAPI$^{-ve}$ (precursor B cell) and CD19$^{+ve}$B220$^{+ve}$IgM$^{+ve}$CD43$^{-ve}$DAPI$^{-ve}$ (immature/mature B cell). Sorting was performed on a FACS-VantageSE cell sorter (BD Biosciences), and DAPI$^{+ve}$ dead cells were gated out.

Progenitor B cells were grown in IMDM (Invitrogen) with 10% heat inactivated FBS, $5 \times 10^{-5}$ M 2-mercaptoethanol (Sigma), and 50 U/ml penicillin and 50 μg/ml streptomycin (Lonza).

To assay proliferation, FACS-sorted $1 \times 10^5$ progenitor B cells were cultured in each well of 96-well plates for 48 h with 10 ng/ml IL-7. On day 2, 0.5 μCi of [3H]Thymidine (Amersham, UK) was added for additional 18 h. [3H]Thymidine incorporation was determined on a liquid scintillation counter (Topcount, Perkin-Elmer) after harvesting (Tomtec cell harvester).

For immunoblotting, progenitor B cells were cultured for 6 days with IL-7. IL-7 was then withdrawn for 4 h and the cells stimulated with 30 μg/ml anti-Igβ (HM79-11; AbD serotec) for the indicated times. Lysates were run on 10% polyacrylamide gels and immunoblotting carried out using standard techniques. Antibodies used are listed in Supplementary Table 1.

### Transduction of retrovirus

The cloning of retroviral vectors for Bcl2A1 and Pax5 and the production of viruses is described in Supplementary data. $1 \times 10^6$ FACS-sorted B220$^{+ve}$CD43$^{high}$IgM$^{-ve}$DAPI$^{-ve}$ B cells were cultured in a 24-well plate for 48 h with 10 ng/ml IL-7. On day 2, retrovirus is added to these cells by spinoculation using 10 μg/ml polybrene. After centrifugation for 90 min at room temperature, cells were kept in an incubator for 1–3 h. Then, these cells were replated with fresh medium containing 10 ng/ml IL-7 in 48-well plates for 48 h. On day 4, cells were harvested into 48-well plates with fresh medium containing no IL-7. On day 6, transduced cells were stained with surface markers and analysed either on FACS-Canto or Fortessa instruments. For cell-cycle analysis of transfected cultures, cells were loaded for 30 min at 37°C with Hoechst at 5 μg/ml in media with 1% FCS. Cells were then resuspended in PBS + 1%FCS, Fc blocked and stained with B220-APC/Cy7, CD19-PerCP/Cy5.5, IgM-PE/Cy7 and CD43-APC.

### Flow cytometry

Cells were stained with combinations of fluorochrome-labelled Abs according to standard procedures for measuring the expression of surface or intracellular markers on FACS-Canto or FACS-Calibur flow cytometers (BD Biosciences). For measuring intracellular protein expression, cells were fixed and permeabilized using a BD Biosciences perm/fix kit. Cell cycle of IL-7 stimulated progenitor B cells was analysed with propidium iodide (Sigma) staining after ethanol fixation. Apoptotic cells were determined using either propidium iodide or annexin V staining (BD Annexin kit). FACS data were analysed on Flowjo software. Lin$^{-ve}$ cells were determined by a lack of staining for CD3, B220, pan NK, Gr1, CD11b, TCRλ/δ, CD4 and CD8. Additional controls for the antibodies used to detect intracellular proteins are shown in Supplementary Figure 13.

### BrdU uptake

Mice received 1 mg BrdU (BD Biosciences) in 100 μl PBS intraperitoneally and were sacrificed after 3.25 h. Bone marrow was harvested and stained for appropriate surface markers and both BrdU incorporation and 7-AAD were determined by manufacturer's protocol of FITC BrdU Flow kit (BD Biosceinces) and analysed by FACS.

### V-DJ rearrangement

V-DJ rearrangement analysis was performed as described (Perlot et al, 2010). Briefly, genomic DNA was extracted from FACS-sorted progenitor or precursor B cells and five-fold dilutions (starting from 50 ng) were used for PCR analysis of V$_H$-DJ$_H$ and V$_k$-J $_k$ rearrangement. PCR products were resolved on 1% agarose/TAE gel, denatured for 20 min in 1.5 M NaCl, 0.5 M NaOH, neutralized with 3 M NaCl, 0.5 M Tris–HCl (pH 7) for 20 min and transferred onto HyBond-N+ nitrocellulose membrane (Amersham) for 16 h in 20× SSC Buffer (3 M NaCl, 0.3 M Na-citrate). Membranes were washed in 2× SSC for 20 min, fixed by baking at 80°C for 2 h in an oven and blocked in hybridization solution (0.72 M NaCl, 0.09 M Na-citrate, 4 mM EDTA, 0.1% Polyvinylpyrrolidone, 0.1% Ficoll, 0.1% BSA, 0.1% SDS and 100 μg/ml denatured Salmon Sperm DNA) for 1 h. Alexa680-conjugated pro-Be was diluted into the hybridization solution at 5 ng/ml and incubated with the membranes at 60°C for 16 h. Membranes were washed in hybridization solution at 60°C and analysed using LICOR system. The germline and recombined mRNA levels of heavy and light chain were measured on real-time PCR using primers described (Fuxa et al, 2004). FACS-sorted precursor B-cell genomic DNA was analysed on real-time PCR for lambda chain rearrangement using primers as described (Lukin et al, 2010).

### Real-time PCR

Total RNA from FACS-sorted B-cell subsets was extracted using the RNeasy micro kit (Qiagen) and reverse-transcribed using iScript (Bio-Rad). qPCR was carried out using SYBRgreen-based detection and the primers sets are listed in Supplementary Table 2. The expression level of the gene of interest was determined relative to the expression of HPRT.

### Supplementary data

Supplementary data are available at The EMBO Journal Online (http://www.embojournal.org).

## Acknowledgements

We thank Dario Alessi (University of Dundee) and Dimitris Kioussis (NIMR) for mouse strains and Arlene Whigham and Marouan Zarrouk for technical assistance. We thank MRC-Protein Phosphorylation Unit (PPU) DNA Sequencing Service (coordinated by Nicholas Helps) for DNA sequencing. This work was supported by the Medical Research Council, Arthritis Research UK, Wellcome Trust (0181867/Z/06/Z) and the pharmaceutical companies supporting the Division of Signal Transduction Therapy Unit (AstraZeneca, Boehringer-Ingelheim, GlaxoSmithKline, Merck KgaA, Janssen Pharmaceutica and Pfizer).

Author contributions: RKCV was responsible for the analysis and majority of the experimental work. RKCV and VM carried out the retroviral rescue experiments. RC carried out cell sorting and contributed to the analysis of FACS data, JCP was responsible for the histology, VAM, PCM and AN contributed to the biochemical analysis and FS and LS assisted with in vivo experiments. RT generated all the DNA clones used in the study. RKCV and JSCA coordinated the study and wrote the manuscript.

## Conflict of interest

The authors declare that they have no conflict of interest.

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
