## [Review Process File · The EMBO Journal]

Manuscript EMBO-2012-82893

PDK1 regulates VDJ recombination, cell cycle exit and survival during B cell development.

Ram K.C. Venigalla, Victoria A McGuire, Rosemary Clarke, Janet C.Patterson-Kane, Ayaz Najafov, Rachel Toth, Pierre C. McCarthy, Frederick Simeons, Laste Stojanovski and J. Simon C. Arthur

Corresponding author: Simon Arthur, University of Dundee

Review timeline:

Submission date:	05 August 2012
Editorial Decision:	03 September 2012
Revision received:	18 January 2013
Accepted:	30 January 2013

Transaction Report:

Editor: Karin Dumstrei

1st Editorial Decision

03 September 2012

Thank you for submitting your manuscript to the EMBO Journal. Your study has now been seen by three referees and their comments are provided below.

As you can see, the referees find the analysis interesting and suitable for publication in the EMBO Journal. They raise a number of specific concerns that I anticipate you should be able to address. Given the comments provided, I would like to invite you to submit a suitably revised manuscript for our consideration.

When preparing your letter of response to the referees' comments, please bear in mind that this will form part of the Review Process File, and will therefore be available online to the community. For more details on our Transparent Editorial Process, please visit our website: <http://www.nature.com/emboj/about/process.html>

Thank you for the opportunity to consider your work for publication. I look forward to your revision.

REFEREE REPORTS

Referee #1

General Comments

The authors have described a novel role for PDK1 of Ig loci recombination, cell cycle exit and survival of early B-cell progenitors. Use of conditional ablation of PDK1 in B-cells, resulted in a severe developmental block at the pro-B to pre-B cell stages. PDK1^{-/-} pro-B cells displayed defective VDJ recombination of Ig heavy chain loci with a major defect in the recombination of DJ segments to distal V (J558). Impaired recombination correlated with a decrease in Pax5 expression and its target genes IRF4, IRF8, and Aiolos, which also resulted in an impairment of pre-B cells to rearrange Ig light chain and exit cell cycle leading to an increase in apoptosis and a profound decrease in pre-B cell numbers. These defects can be supplemented by exogenous expression of Pax5 and the anti-apoptotic molecule, Bcl2A1, to induce full differentiation of PDK1^{-/-} B-cells into immature B-cells. In general, the authors have provided substantial evidence regarding the importance of PDK1 in B lymphopoiesis. Mechanistic insights are provided as well. However, a few issues need to be addressed before a cohesive and integrated understanding of how PDK contributes to B lymphopoiesis can be arrived at.

Major concerns:

1. Full controls are not provided for the experiments depicted in Figs. 8C-E. Does Bcl2A1 and Pax5 really complement deficiencies or do they, in general, augment B lymphopoiesis when over-expressed. What happens when these molecules are expressed in WT cells?
2. The Pax5 reconstitutions would be more compelling if the authors demonstrate that ectopic Pax5 expression restored distal Vh recombination which has been previously shown to a specific Pax5 function.
3. The authors waffle quite a bit on the relative contributions of enhanced apoptosis or aberrant proliferation to the phenotype. Most of the data clearly favors apoptosis as the major contributor. They need to just state this. They also need to be more thoughtful about the BrdU experiment.

Minor concerns:

1. There are some inconsistencies between mRNA changes (modest) and protein levels (more severe, example $\lambda 5$). This could be due to the known effects of PI3K/Akt on protein translation. This should be discussed.
2. Likewise, some of the phenotype could be due to aberrant positioning in the BM as PDK1 affects the expression of chemokines and adhesion molecules (Finlay, D.K. 2009, JEM). This should be discussed.
3. The quoted Akt1/2 KO paper used fetal liver. This could account for the discrepancies between this paper and most reports focused on PI3K. The authors' data matches more with the latter. This should be discussed.
4. Recent data indicates that PI3K/Akt is downstream of the IL-7R and not the pre-BCR (Nat Immunol, 13:00). This needs to be considered in the discussion and in interpreting the results of experiments in which Ig-beta is artificially cross-linked on pro-B cells.
5. There are several grammatical errors in the manuscript and the labeling of the figures is inconsistent.

Referee #2

Venigalla et al., PDK1 regulate VDJ recombination, cell cycle exit and survival during B cell development

Summary:

In this manuscript, the authors investigate the role of PDK-1 in the development of the immune system, specifically early B cell development. This is an important question in lymphocyte development as many of the effector kinases, which are known targets of PDK-1, have roles in early lymphocyte development; but the role of PDK-1 in B cell lymphopoiesis is unknown. The authors use a conditional knock-out in combination with a Vav-cre expression to generate mice deficient in PDK-1 in early hematopoiesis. While they clearly demonstrate that PDK-1 is deleted in bone marrow cells, they do not demonstrate at defined developmental stage at which PDK-1 is lost. Nevertheless, the data is compelling and supports a role for PDK-1 in the regulation of early lymphocyte survival, proliferation, and V(D)J recombination. PDK-1 deficient mice have defects in transition from pro-B to pre-B cell stages and a complete block in development beyond the pre-B cell stage. The authors conclude this is secondary to reduced viability and deficient cell cycle arrest of pre-B cells.

Primary Concerns:

1. The authors focus their studies principally on signaling in the absence of IL-7, which supports a role for PDK-1 downstream of pre-BCR. However, Akt (and possibly some of the other kinases) also signal downstream of IL-7. In figure 6F and 6G, the authors demonstrate abnormal cell responses to IL-7 in vitro with increased cell death and altered proliferation. However, they do not discuss how PDK-1 could be involved in IL-7 signaling and the changes in kinase activation in Fig. 5B are only done under conditions of IL-7 withdrawal. In Fig. 5B, Foxo1 phosphorylation is already low at time 0 which suggests a possible defect in IL-7 to Akt signaling. Is there a defect in IL-7 signaling through PDK-1 to Akt and other kinases?
2. The authors use in vitro IL-7 cultures to examine the differentiation of B cells with and without PDK-1 (Fig. 5B, 6A, 6E, 6F, 6G). They use this model to conclude that pro-B cell proliferation is defective and that maturation is defective. It's important to note that pro-B cells normally rapidly mature in this culture system to become pre-B cells in the presence of IL-7 and then to immature B cells (to a certain extent in IL-7 but to a greater extent after removal of IL-7). The authors should demonstrate the phenotype of the cells throughout the IL-7 culture and particularly at the points of assessment for the other studies (proliferation, survival, maturation, etc). If the culture is primarily pre-B cells during their proliferation or survival studies, this could change their interpretations of this data. This is particularly relevant given the results that pro-B cell survival is defective in vitro (Fig. 6F) but is normal in vivo (Fig. 8A).

Minor concerns:

1. All graphs need p-values, not just the ones with statistically significant differences. Examples: Fig. 1C, 1E, 3A, 3C, 4C, 4D, 5A, 6D, 6F, 6G, 7C, Suppl. Fig. 1C Suppl. Fig. 3, Suppl. Fig. 10C
2. Supplementary Fig. 4 has flow cytometry on lymph nodes. Total lymph node cell numbers should be included as well.
3. Add number of mice used to bar graphs. Fig 1C, 1E, 3A, 3C,
4. Total number of B cells in bone marrow should be added to Fig. 3C.
5. In Figure 4, for IgH rearrangement, it is unclear if the IgH locus is equally accessible in the wild-type and PDK1-deficient cells. If IgH germline transcription is decreased (similar to the decrease seen in the IgL locus), this would explain the reduction in IgH rearrangements seen. This is particularly important as no explanation for reduced IgH rearrangement is provided by the other data in the paper.
6. For Figure 5, total RSK2 levels are not shown but are relevant to determination of RSK2 phosphorylation levels. Additionally, is the ERK1/2 phosphorylation of RSK2 normal and the only defect is PDK-1 phosphorylation of RSK2
7. Figure 5 legend needs to be corrected to reflect the figure.
8. The authors mention that Bcl2A1 is an NF κ B-regulated gene and that the PDK-1 deficient cells

also have reduced I κ B α expression, which then remark is consistent with reduced NF κ B activity. If NF κ B activity is affected by PDK-1 activity, this should be demonstrated by EMSA or nuclear localization of the p50/p65 complex. This would be a more convincing method to demonstrate effects on NF κ B activity.

9. In figure 8, expression of Pax5 affects IgM expression and recues, in combination with Bcl2A1, B cell maturation. Does expression of Pax5 affect the cell cycle in PDK-1-deficient cells? If it's responsible for Aiolos expression and CCND3 downregulation, it would be expected that expression of Pax5 re-activates cell cycle regulation in PDK-1-deficient cells.

10. In Figure 8, the correction of cell death by Bcl2A1 is shown only by FSC v. SSC. Annexin staining should be included here as more accurate assessment of cell viability.

Referee #3

The manuscript of Venigalla et al addresses the role of the kinase PDK1 in B cells. They clearly show that PDK1 is required for the transition from the pro-B to pre-B cell stage. The defect is cell autonomous and is characterised by impaired V to DJ recombination of IgH, predominantly at the distal VHJ558 locus, decreased survival (decreased Mcl1 and A1 expression) and impaired pre-BCR signalling. mRNA analysis suggests that while Pax5 expression was normal in pro-B cells, expression was reduced in cKO pre-B cells. In keeping with reduced Pax5 activity, the cKO pre-B cells had decreased IRF4, IRF8 and Aiolos, three transcription factors important at the pre-B cell stage that are known Pax5 activated genes. Finally rescue experiments show that a combination of blocking apoptosis through ectopic A1 and overexpression of Pax5 worked in synergy to "rescue" B cell maturation in vitro.

Overall this is a very clear and well written and presented paper that provides some important insights into the role of the pre-BCR in early B cell development. I feel that the manuscript would be acceptable to publish in EMBO J if the authors address the following issues.

1. Intracellular FACS for B cell transcription factors (figure 7D). This panel needs some controls and validation as with the exception of Irf4, none of these TFs are routinely assayed in this manner in the field (I was unable to check the Ab details as suppl table 1 was missing in my pdfs). One necessary control is a background stain to see how specific the stains are. One concern is that for Pax5, the mRNA is not differentially expressed in pro-B cells (figure 4D), but the FACS in figure 7D shows otherwise. Conversely, IRF4 is not differentially expressed in pre-B cells on an RNA level, but is via FACS. I feel that at least for Pax5 and preferably for IRF4, 8 and Aiolos the results should be confirmed by Western blots on pre-B cells, as they are crucial to the major conclusions of the paper.
2. The rescue experiments in figure 8C-E are not entirely convincing. Firstly the rationale for gating only IgM⁺ IgD⁺ and not all IgM⁺ cells is not provided. The major biologically relevant rescue is to get IgM⁺ B cells (the authors could also stain for IgK if they are worried about the low expression of the preBCR). IgD is just a splice variant of IgM that is expressed later on mature B cells. The authors need to provide some collated data to confirm the reproducibility and statistical significance of the proportion of IgM⁺ IgD⁺ and IgM⁺ IgD⁻ cells carrying each retrovirus. To authors should confirm that the rescue with Pax5 result from a rescue of physiological Pax5 expression levels.
3. A minor point is that the cell number deriving from the CLP staining and gating in figure 3B is probably not reliable as it appears that the expanded myeloid progenitors (kit⁺sca⁻) are falling into the CLP gate. The CLP plots could be improved by using Flt3, as is now standard in this strategy, but the easier option would be to remove figure 3B entirely as it is clear that the subsequent stages of B cell development are produced normally.
4. Suppl table1 is missing from at least my files.

Referee #1

General Comments

The authors have described a novel role for PDK1 of Ig loci recombination, cell cycle exit and survival of early B-cell progenitors. Use of conditional ablation of PDK1 in B-cells, resulted in a severe developmental block at the pro-B to pre-B cell stages. PDK1^{-/-} pro-B cells displayed defective VDJ recombination of Ig heavy chain loci with a major defect in the recombination of DJ segments to distal V (J558). Impaired recombination correlated with a decrease in Pax5 expression and its target genes IRF4, IRF8, and Aiolos, which also resulted in an impairment of pre-B cells to rearrange Ig light chain and exit cell cycle leading to an increase in apoptosis and a profound decrease in pre-B cell numbers. These defects can be supplemented by exogenous expression of Pax5 and the anti-apoptotic molecule, Bcl2A1, to induce full differentiation of PDK1^{-/-} B-cells into immature B-cells. In general, the authors have provided substantial evidence regarding the importance of PDK1 in B lymphopoiesis. Mechanistic insights are provided as well. However, a few issues need to be addressed before a cohesive and integrated understanding of how PDK contributes to B lymphopoiesis can be arrived at.

Major concerns:

1. Full controls are not provided for the experiments depicted in Figs. 8C-E. Does Bcl2A1 and Pax5 really complement deficiencies or do they, in general, augment B lymphopoiesis when over-expressed. What happens when these molecules are expressed in WT cells?

We have added some further experiments to address this point. The overexpression of Bcl2A1 in wild type cells does not have an effect on the differentiation of pro-B cells in the in vitro culture system we have used. Consistent with its role for promoting B cell development Pax5 overexpression in wild type cells does result in a trend for an increase in IgM expression in GFP-Pax5^{+ve} cells compared to GFP-Pax5^{-ve} cells (this data is now in supplementary figure 12). This effect was more apparent when Bcl2A1 was expressed in combination with Pax5; this is most likely because on its own Pax5 overexpression resulted in decreased cell survival or proliferation in wild type cells resulting in a very low percentage of GFP-Pax5^{+ve} cells. The increase in the percentage of either IgM^{+ve} or IgM^{+ve}/IgD^{+ve} cells caused by a combination of Bcl2A1 and Pax5 expression was however less in wild type cells than that seen in PDK1 knockout cells.

2. The Pax5 reconstitutions would be more compelling if the authors demonstrate that

ectopic Pax5 expression restored distal Vh recombination that has been previously shown to a specific Pax5 function.

While we agree that this would be the predicted result, we have not been able to examine this. When we transfect Pax5 into PDK1 knockout B cells, we do not see high rates of transfection in the surviving cells. While we see sufficient Pax5⁺ cells for analysis based on FACS, when we can gate out untransfected cells, there is not a sufficient number of Pax5⁺ cells to allow us generate meaningful data for a PCR based analysis of recombination in the total cell population. We would therefore need to carry out further cell sorting on the transfected population. Even starting with 8 to 10 mice for one transfection we do not get sufficient Pax5⁺ cells surviving after the retroviral transfection for us make it viable for us to resort the transfected cells with the experimental setups we currently have.

3. The authors waffle quite a bit on the relative contributions of enhanced apoptosis or aberrant proliferation to the phenotype. Most of the data clearly favors apoptosis as the major contributor. They need to just state this. They also need to be more thoughtful about the BrdU experiment.

We have revised the text with this in mind.

Minor concerns:

1. There are some inconsistencies between mRNA changes (modest) and protein levels (more severe, example $\lambda 5$). This could be due to the known effects of PI3K/Akt on protein translation. This should be discussed.

We agree with the reviewer on this point, and have added some discussion of this in the text.

2. Likewise, some of the phenotype could be due to aberrant positioning in the BM as PDK1 affects the expression of chemokines and adhesion molecules (Finlay, D.K. 2009, JEM). This should be discussed.

We have included some discussion of this in the revised paper.

3. The quoted Akt1/2 KO paper used fetal liver. This could account for the discrepancies between this paper and most reports focused on PI3K. The authors' data matches more with the latter. This should be discussed.

We have included some discussion of this in the revised paper. In addition to the point raised by the reviewer, there is also a possibility of compensation from upregulation of Atk3 in the Akt1/2 paper which we have also referred to.

4. Recent data indicates that PI3K/Akt is downstream of the IL-7R and not the pre-BCR (Nat Immunol, 13:00). This needs to be considered in the discussion and in interpreting the results of experiments in which Ig-beta is artificially cross-linked on pro-B cells.

We have included some discussion of this point. Our intention was not to suggest that the PDK1 did not function downstream of IL-7 *in vivo*. While our data shows a clear role for PDK1 in regulating B cell development in the bone marrow, we cannot from our data demonstrate if IL-7, the pre-BCR or both stimuli are the upstream signals for PDK1 and

Akt in this process. We have added some discussion of the recent Nature Immunology paper on the relative importance of IL-7 and pre-BCR signaling for Akt activation to the revised manuscript.

For B cell development to occur both the pre-BCR and IL-7 signaling is required and hence we need to show that these receptors were expressed in the knockout cells. Our aim with data in Fig 5C however was only to confirm that PDK1 was required for the activation of Akt in B cells. We did not mean to imply from this that the pre-BCR was the critical Akt activating signal *in vivo*. We would expect similar results for PDK1 knockout with a range of stimuli and Ig- β cross-linking was presented as it gave a robust activation of Akt *in vitro*. We have added some experiments with a recently described PDK1 inhibitor showing that it blocks Akt activation in the 70z/3 pre-B cell line in response to IL-7 (supplementary figure 9). We have also revised the text to clarify these points.

5. *There are several grammatical errors in the manuscript and the labeling of the figures is inconsistent.*

We have tried to find and fix these issues.

Referee #2

Venigalla et al., PDK1 regulate VDJ recombination, cell cycle exit and survival during B cell development

Summary:

In this manuscript, the authors investigate the role of PDK-1 in the development of the immune system, specifically early B cell development. This is an important question in lymphocyte development as many of the effector kinases, which are known targets of PDK-1, have roles in early lymphocyte development; but the role of PDK-1 in B cell lymphopoiesis is unknown. The authors use a conditional knock-out in combination with a Vav-cre expression to generate mice deficient in PDK-1 in early hematopoiesis. While they clearly demonstrate that PDK-1 is deleted in bone marrow cells, they do not demonstrate at defined developmental stage at which PDK-1 is loss. Nevertheless, the data is compelling and supports a role for PDK-1 in the regulation of early lymphocyte survival, proliferation, and V(D)J recombination. PDK-1 deficient mice have defects in transition from pro-B to pre-B cell stages and a complete block in development beyond the pre-B cell stage. The authors conclude this is secondary to reduced viability and deficient cell cycle arrest of pre-B cells.

Primary Concerns:

1. *The authors focus their studies principally on signaling in the absence of IL-7, which supports a role for PDK-1 downstream of pre-BCR. However, Akt (and possibly some of the other kinases) also signal downstream of IL-7. In figure 6F and 6G, the authors demonstrate abnormal cell responses to IL-7 in vitro with increased cell death and altered proliferation. However, they do not discuss how PDK-1 could be involved in IL-7 signaling and the changes in kinase activation in Fig. 5B are only done under conditions of IL-7 withdrawal. In Fig. 5B, Foxo1 phosphorylation is already low at time 0 which suggests a possible defect in IL-7 to Akt signaling. Is there a defect in IL-7 signaling through PDK-1 to Akt and other kinases?*

As discussed in the response to point 4 above, we did not mean to imply that PDK1 only acted only downstream of the pre-BCR and not IL-7 or other stimuli. We have therefore revised the results and discussion to make this point clear. We used pre-BCR signaling as an example of a stimulus that activates Akt to show the requirement for PDK1 occurred in B cell progenitors. We agree that PDK1 would also act downstream of IL-7 to activate Akt and have shown that this occurs in the 70Z/3 pre-B cell line (supplementary figure 9). To our knowledge no other kinase apart from PDK1 has been shown to phosphorylate Thr308 in Akt in any cell system tested, and it therefore seems very likely that PDK1 would fulfill this role downstream of all Akt activating stimuli in B cells.

2. The authors use in vitro IL-7 cultures to examine the differentiation of B cells with and without PDK-1 (Fig. 5B, 6A, 6E, 6F, 6G). They use this model to conclude that pro-B cell proliferation is defective and that maturation is defective. It's important to note that pro-B cells normally rapidly mature in this culture system to become pre-B cells in the presence of IL-7 and then to immature B cells (to a certain extent in IL-7 but to a greater extent after removal of IL-7). The authors should demonstrate the phenotype of the cells throughout the IL-7 culture and particularly at the points of assessment for the other studies (proliferation, survival, maturation, etc). If the culture is primarily pre-B cells during their proliferation or survival studies, this could change their interpretations of this data. This is particularly relevant given the results that pro-B cell survival is defective in vitro (Fig. 6F) but is normal in vivo (Fig. 8A).

We agree with the reviewer that the wild type pro-B cells differentiate in IL-7 cultures to become pre-B cells before upregulating the expression of the BCR. We have revised the text in the results to make this point clear. The PDK1 knockout cells fail to develop into pre-B cells or IgM+ve cells in the *in vitro* culture system, which is consistent with what is observed *in vivo*. In cell culture models, even with IL-7 present, the cells do not receive the same degree of trophic support that they receive *in vivo*; this is most likely why we see increased cell death in the PDK1 knockout pro-B cells in the cultures but not *in vivo*. We have commented on this in the discussion. In this respect it worth noting that wild type cells also show high levels cell death in the *in vitro* culture system.

In order to get efficient downregulation of CD43 (which occurs as cells passage from pro-B to pre-B) we need to re-express both Bcl2A1 and Pax5 (experiments to show this have been added to supplementary figure 12). Expression of Bcl2A1 and Pax5 also promotes the transition of the PDK1 knockout cells from the G2/M phase of the cell cycle to the G1/0 phase, which would be consistent with the cells then becoming competent to rearrange their light chains (cell cycle data has been added to figure 8 and supplementary figure 12).

Minor concerns:

1. All graphs need p-values, not just the ones with statistically significant differences. Examples: Fig. 1C, 1E, 3A, 3C, 4C, 4D, 5A, 6D, 6F, 6G, 7C, Suppl. Fig. 1C Suppl. Fig. 3, Suppl. Fig. 10C

These have been added.

2. Supplementary Fig. 4 has flow cytometry on lymph nodes. Total lymph node cell numbers should be included as well.

This has been added.

3. Add number of mice used to bar graphs. Fig 1C, 1E, 3A, 3C,

These numbers have been included in the figure legends.

4. Total number of B cells in bone marrow should be added to Fig. 3C.

While there was a trend for a decrease in the total B cells in the knockout this did not reach statistical significance ($p > 0.05$). A sentence to indicate this has been added to the text. We feel the more informative data is the absolute numbers of the B cells subsets and therefore have focused on this in the actual figure.

5. In Figure 4, for IgH rearrangement, it is unclear if the IgH locus is equally accessible in the wild-type and PDK1-deficient cells. If IgH germline transcription is decreased (similar to the decrease seen in the IgL locus), this would explain the reduction in IgH rearrangements seen. This is particularly important as no explanation for reduced IgH rearrangement is provided by the other data in the paper.

This data has been added to figure 4. The germline transcription of the IgH locus is not significantly effected by the knockout of PDK1. One potential explanation, as mentioned in the discussion, is the decrease in Ligase IV expression, however unfortunately our data does not pinpoint the precise molecular mechanism behind this.

6. For Figure 5, total RSK2 levels are not shown but are relevant to determination of RSK2 phosphorylation levels. Additionally, is the ERK1/2 phosphorylation of RSK2 normal and the only defect is PDK-1 phosphorylation of RSK2

We have previously shown that PDK1 knockout does not affect the phosphorylation of RSK2 by ERK1/2 and does not affect RSK2 expression (Curr Biol, 2000, 10:439-48 and EMBO J. 2003, 15:4202-11). While we did attempt to look at the phosphorylation of RSK2 by ERK1/2 in the pre-B cells, the phospho antibody for this site was not sensitive enough to detect the levels of endogenous RSK2 expressed in these cells.

7. Figure 5 legend needs to be corrected to reflect the figure.

This has been corrected.

8. The authors mention that Bcl2A1 is an NFkB-regulated gene and that the PDK-1 deficient cells also have reduced Ikbα expression, which then remark is consistent with reduced NFkB activity. If NFkB activity is affected by PDK-1 activity, this should be demonstrated by EMSA or nuclear localization of the p50/p65 complex. This would be a more convincing method to demonstrate effects on NFkB activity.

We mentioned this possibility in the paper as it has previously been reported that PDK1 may regulate NFkB and Bcl2A1 has been linked to NFkB. How Bcl2A1 transcription is controlled is an interesting question and not one that has been extensively studied. While we wish to address this in future, we felt that this is beyond the scope of our current study, as we would not be able to adequately address this question given the constraints in terms of space as well as the time for the current study. It would represent

a whole paper in its own right. In addition we are limited by the numbers of knockout mice available. To generate sufficient pro-B cells for the EMSA would take a significant number of mice and on its own would not confirm that Bcl2A1 is regulated by NFkB in B pro-B cells.

9. In figure 8, expression of Pax5 affects IgM expression and recues, in combination with Bcl2A1, B cell maturation. Does expression of Pax5 affect the cell cycle in PDK-1-deficient cells? If it's responsible for Aiolos expression and CCND3 downregulation, it would be expected that expression of Pax5 re-activates cell cycle regulation in PDK-1-deficient cells.

We have carried out some further experiments to look at this, and in agreement with this prediction the expression of Pax5 in the knockout cells results in exit from the G2/M phase of the cell cycle, which would be required for the initiation of light chain recombination. For this experiment we infected cells with both Pax5 and Bcl2A1 retroviruses, this was because the on its own the Pax5 virus gives very low numbers of GFP⁺ve cells which would have made the cell cycle analysis more difficult. Comparison of the Bcl2A1⁺ve and ⁻ve cells showed that Bcl2A1 on its own did not affect the cell cycle profile. This data is shown in Figure 8 and supplementary figure 12.

10. In Figure 8, the correction of cell death by Bcl2A1 is shown only by FSC v. SSC. Annexin staining should be included here as more accurate assessment of cell viability.

We have confirmed that Bcl2A1 re-expression increased cell survival using DAPI staining. This data has been added to supplementary figure 12. The correlation between the FSC v. SSC analysis and the DAPI staining was very strong. As we had a larger n number for the FSC v. SSC analysis we left this in the main figure and added the DAPI to the supplementary figure.

Referee #3

The manuscript of Venigalla et al addresses the role of the kinase PDK1 in B cells. They clearly show that PDK1 is required for the transition from the pro-B to pre-B cell stage. The defect is cell autonomous and is characterized by impaired V to DJ recombination of IgH, predominantly at the distal VHJ558 locus, decreased survival (decreased Mcl1 and A1 expression) and impaired pre-BCR signaling. mRNA analysis suggests that while Pax5 expression was normal in pro-B cells, expression was reduced in cKO pre-B cells. In keeping with reduced Pax5 activity, the cKO pre-B cells had decreased IRF4, IRF8 and Aiolos, three transcription factors important at the pre-B cell stage that are known Pax5 activated genes. Finally rescue experiments show that a combination of blocking apoptosis through ectopic A1 and overexpression of Pax5 worked in synergy to "rescue" B cell maturation in vitro.

Overall this is a very clear and well written and presented paper that provides some important insights into the role of the pre-BCR in early B cell development. I feel that the manuscript would be acceptable to publish in EMBO J if the authors address the following issues.

1. Intracellular FACS for B cell transcription factors (figure 7D). This panel needs some controls and validation as with the exception of Irf4, none of these TFs are routinely

assayed in this manner in the field (I was unable to check the Ab details as suppl table 1 was missing in my pdfs). One necessary control is a background stain to see how specific the stains are. One concern is that for Pax5, the mRNA is not differentially expressed in pro-B cells (figure 4D), but the FACS in figure 7D shows otherwise. Conversely, IRF4 is not differentially expressed in pre-B cells on an RNA level, but is via FACS. I feel that at least for Pax5 and preferably for IRF4, 8 and Aiolos the results should be confirmed by Western blots on pre-B cells, as they are crucial to the major conclusions of the paper.

We have added supplemental data to show the background staining for the pre-B for the intracellular antibodies used (supplementary figure 13). We have also expressed the relevant proteins in HEK-293 cells and shown that the antibodies can recognize the appropriate overexpressed protein. In addition for Pax5, we have obtained similar results with two independent Pax5 antibodies (1H9 clone eBioscience and 24/ clone, BD Biosciences). The 1H9 clone have been used previously in the literature (PMID:21606506)

2. The rescue experiments in figure 8C-E are not entirely convincing. Firstly the rationale for gating only IgM⁺ IgD⁺ and not all IgM⁺ cells is not provided. The major biologically relevant rescue is to get IgM⁺ B cells (the authors could also stain for IgK if they are worried about the low expression of the pre-BCR). IgD is just a splice variant of IgM that is expressed later on mature B cells. The authors need to provide some collated data to confirm the reproducibility and statistical significance of the proportion of IgM⁺ IgD⁺ and IgM⁺ IgD⁻ cells carrying each retrovirus. To authors should confirm that the rescue with Pax5 result from a rescue of physiological Pax5 expression levels.

We have now included data for to show the reproducibility of this data (revised fig 8). We agree with the reviewer that the relevant rescue is the generation of IgM positive cells. The re-expression of Pax5 and Bcl2A1 in the PDK1 knockout cells does result an increase in the number of cells that are positive IgM and this data is quantified in supplementary figure 12. As shown by the FACS plots for IgM, the number of cells that are rescued however is a small percentage of the total, and this makes the start of the IgM⁺ gate more subjective. For this reason we chose to leave the analysis for IgM/IgD positive cells in the main figure as we felt that this makes the gating more robust.

The low percentage of GFP-Pax5 cells when we transfect either wild type or PDK1 knockout cells makes it difficult to look at the expression level by immunoblotting. When we transfect GFP-Pax5 into wild type pro-B cells and look at the levels of Pax5 protein by intracellular staining (which detects both endogenous and GFP-Pax5) then the GFP⁺ cells only express slightly higher levels of Pax5 relative to wild type cells indicating that expression level of the GFP-Pax5 is likely to be similar to the endogenous levels. The group of cells GFP⁺ cells that express lower levels of Pax5, based on shifts

in their forward/side scatter, cells which are starting to die. This is in line with the ability of Pax5 overexpression to induce cell death.

3. A minor point is that the cell number deriving from the CLP staining and gating in figure 3B is probably not reliable as it appears that the expanded myeloid progenitors (kit+sca-) are falling into the CLP gate. The CLP plots could be improved by using Flt3, as is now standard in this strategy, but the easier option would be to remove figure 3B entirely as it is clear that the subsequent stages of B cell development are produced normally.

While we agree that further stains would improve the resolution, we have at the moment left this data in as the other two reviewers have not asked for it to be removed. We have however qualified the description of this in the text. If necessary we would be happy to remove this panel.

4. Suppl table1 is missing from at least my files.

This table has been added.

Acceptance letter

30 January 2013

Thank you for submitting your revision to the EMBO Journal. Referees #2 and 3 were available to review the revised version and I have now received their comments.

As you can see below, both referees appreciate the introduced changes. I am therefore very pleased to accept the paper for publication here.

Thank you for contributing to the EMBO Journal

REFeree REPORTS

Referee #2

The authors have addressed all of our comments. This is a very interesting paper that for the first time implicates PDK1 in the regulation of antigen receptor gene assembly and provides mechanistic information about how it does this.

Referee #3

The authors have done an adequate job of the revision and in my opinion the manuscript is ready for publication as is.